# Who Gets Credit or Blame? Attributing Accountability in Modern AI Systems

**Shichang Zhang**[1] **Hongzhe Du**[2] **Jiaqi W. Ma**[3] **Himabindu Lakkaraju**[1]

## Abstract

Modern AI systems are typically developed through multiple stages—pretraining, fine-tuning rounds, and subsequent adaptation or alignment—where each stage builds on the previous ones and updates the model in distinct ways. This raises a critical question of accountability: when a deployed model succeeds or fails, which stage is responsible, and to what extent? We pose the *accountability attribution* problem for tracing model behavior back to specific stages of the model development process. To address this challenge, we propose a general framework that answers counterfactual questions about stage effects: *how would the model's behavior have changed if the updates from a particular stage had not occurred?* Within this framework, we introduce estimators that efficiently quantify stage effects without retraining the model, accounting for both the data and key aspects of model optimization dynamics, including learning rate schedules, momentum, and weight decay. We demonstrate that our approach successfully quantifies the accountability of each stage to the model's behavior. Based on the attribution results, our method can identify stages associated with spurious correlations in image classification and text toxicity detection tasks and guide targeted follow-up interventions. Our approach provides a practical tool for model analysis and represents a significant step toward more accountable AI development.

## 1. Introduction

Modern AI systems typically comprise multiple development stages, including pretraining, domain-specific fine-tuning, and subsequent adaptation or alignment (Lopez-Paz

[1]Harvard University, Cambridge, MA, USA [2]University of California, Los Angeles, Los Angeles, CA, USA [3]University of Illinois Urbana-Champaign, Urbana-Champaign, IL, USA. Correspondence to: Shichang Zhang <shzhang@hbs.edu>.

*Proceedings of the 43rd International Conference on Machine Learning*, Seoul, South Korea. PMLR 306, 2026. Copyright 2026 by the author(s).

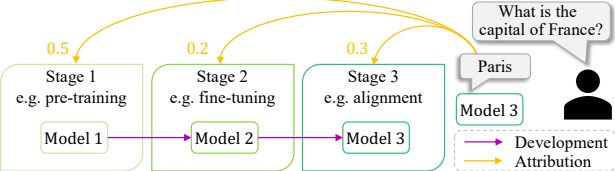

*Figure 1.* Illustration of the **accountability attribution** problem for a generative AI model developed in three stages (e.g., pretraining, fine-tuning, and alignment). The accountability for the output "*Paris*" is attributed to the three stages.

& Ranzato, 2017; Kornblith et al., 2019; Raghu et al., 2019; He et al., 2022; Chen et al., 2020; Radford et al., 2019; Ouyang et al., 2022; Hu et al., 2022). Each stage builds on the previous one to update the model differently, and each stage update involves many steps shaped by both data and optimization dynamics. While this modular structure has become central to achieving good performance, it complicates a critical question of accountability: when a model exhibits harmful, beneficial, or surprising behavior, which development stage bears accountability? This question, lying at the intersection of explainability, causality, and learning dynamics, remains largely underexplored in current practice. As models are increasingly deployed in high-stakes settings, answering this question becomes essential for model debugging, auditing, and enforcing accountability to credit or blame the appropriate stages properly. For instance, determining whether harmful racial biases emerged during pretraining or fine-tuning, identifying which stage introduced undesirable spurious correlations, or understanding whether robustness properties were learned during domain adaptation or the initial training phase.

We formulate the **accountability attribution** problem to address this challenge: tracing model behavior to stages of the model development process that shaped it. This problem relates to, yet remains distinct from, three research directions. First, causal responsibility analysis (Chockler & Halpern, 2004; Halpern & Pearl, 2005; Triantafyllou et al., 2021) provides formal definitions of blame and responsibility through structural causal models but has primarily focused on discrete decision-making settings at small scales (Halpern & Pearl, 2005), making it challenging to apply to high-dimensional, sequential processes like training deep AI models. Second, research on learning dynamics (Ren et al., 2022; Ren & Sutherland, 2025; Park et al.,

2024) investigates how model parameters evolve during training and their consequent impact on test performance, revealing phenomena such as phase transitions (Park et al., 2024). However, this research typically aims at descriptive understanding rather than attributing credit or blame to specific development stages. Third, data attribution (Koh & Liang, 2017; Ghorbani & Zou, 2019; Ilyas et al., 2022; Pruthi et al., 2020; Bae et al., 2024; Wang et al., 2025) traces model behavior to individual data points. While these methods can assign data-level accountability, they primarily study the "average model" expected to be trained from a given dataset, often overlooking the specific model development process (Koh & Liang, 2017). Their assumptions and simplifications (e.g., convexity, permutation invariance) limit their applicability to development stage attribution. Although recent work has extended to be more model-specific (Bae et al., 2024; Wang et al., 2025), they remain data-centric and assume basic SGD optimizers, failing to account for practical complex optimization dynamics.

To address these gaps, we propose a framework for accountability attribution that explicitly analyzes the training process. Our framework builds on the potential outcomes formalism (Rubin, 1974; 2005), enabling counterfactual queries about the effect of model development stages: *how would the model's behavior have changed if the updates from a particular stage had not occurred?* It focuses on estimating the causal effects of stages, which are defined as sets of model update steps determined by both data and optimization dynamics, including influences from learning rate schedules, momentum, and weight decay. This approach provides **model-specific** attribution results by considering the complete model development process. We instantiate this framework using Taylor-approximation-based estimators for the effect of stages. Our estimators are efficient and flexible: they avoid model re-training, scale to deep models, and yield reusable representations that capture the essential influence patterns of each development stage. These stage representations only need to be computed once and can be applied for model behavior on any test input or performance function. We refer to the estimated effect on model performance as the *Accountability Attribution Score (AA-Score)* of a development stage.

Through experiments on vision and language tasks, we show that our method reliably identifies stages that are accountable for critical model behaviors—including the introduction of spurious correlations, the learning of domain generalization, or the degradation from noisy labels. These results position accountability attribution as a practical and principled tool for model analysis and assignment of credit or blame. Our contributions are summarized as follows:

- We pose the **accountability attribution problem** as tracing model behavior to stages of the training process.

- We propose a **general framework** for accountability attribution based on the potential outcomes formalism, enabling counterfactual queries about training stage effects.

- We derive efficient **estimators** within the proposed framework that quantify stage effects while accounting for **optimization dynamics**.

- We demonstrate the framework's **practical utility** across diverse settings, showing that it uncovers influential stages responsible for beneficial and harmful model behaviors.

## 2. Related Work

**Responsibility and causal analysis** The assessment of responsibility is a fundamental challenge in practice that often requires careful consideration of causality (Chockler & Halpern, 2004). Structural causal models serve as powerful tools for formalizing this concept (Halpern & Pearl, 2005; Pearl, 2009), enabling precise definitions of blame and responsibility through counterfactual dependence (Halpern & Kleiman-Weiner, 2018). In the context of AI, these frameworks have been extended to analyze multi-agent settings (Triantafyllou et al., 2021) and human-AI collaboration (Qi et al., 2024). While these formalisms provide valuable perspectives on responsibility attribution, they typically focus on relatively simple problems in small settings with enumerable outcomes, e.g., two people throwing rocks at a bottle. Applying them directly to the high-dimensional, sequential process of training deep AI models presents significant challenges. For such complex processes of AI model training, a notable related work by (Lesci et al., 2024) employs a potential outcome framework to study memorization. While our work shares the goal of using causal reasoning and the potential outcome framework, we focus on attributing model behavior to training stages and determining accountability.

**Learning dynamics** describes how AI models usually learn new knowledge via gradient-based parameter updates. It links changes in the model's parameters or predictions over time, to the gradients generated by learning specific examples (Ren et al., 2022). Through analyzing the learning dynamics, interesting phenomena during training has been explained, such as the "zig-zag" learning path (Ren et al., 2022), the "squeezing effect" of LLM finetuning (Ren & Sutherland, 2025), and the formation of compositional concept spaces (Park et al., 2024). Our work complements these studies by providing a method to quantify the contribution of training stages to the final outcome, helping to explain the mechanisms behind observed dynamic phenomena. Our method is also quantitative and can be efficiently applied to different test data or performance metrics.

**Data attribution** aims to trace model behavior back to the training data instances. Approaches like influence functions, game theory, and repeated retraining are proposed to at-

tribute model behavior to data points (Koh & Liang, 2017; Ghorbani & Zou, 2019; Ilyas et al., 2022; Wang & Jia, 2023; Ley et al., 2024; Deng et al., 2025b). Deng et al. (2025a) provides a detailed survey. Most of these methods study an "average model" expected to be trained from a given dataset and thus are limited for analyzing specific model instances. Moreover, they often rely on assumptions such as convergence, convexity, or permutation invariance of training data that limit their applicability to multi-stage training processes in practice. One line of data attribution research examines specific model development processes, including SGD-Influence (Hara et al., 2019), TracIn (Pruthi et al., 2020), unrolled differentiation (Bae et al., 2024), and DVEmb (Wang et al., 2025). These process-based approaches better capture temporal dependencies by tracing influence along the model update trajectory. The most closely related work to ours is DVEmb, which traces training example influence along the update trajectory using first-order approximations for leave-one-out (LOO) counterfactuals. Our work differs by analyzing the specific counterfactual of development stages instead of only data points, and considering a more complete, practical optimization process incorporating learning rate schedules, momentum, and weight decay.

## 3. Preliminaries

### 3.1. Optimization Dynamics

Deep learning AI models are developed through a sequence of optimization steps for updating model parameters. Let $p(\boldsymbol{x}; \boldsymbol{\theta})$ be a model on instances $\boldsymbol{x} \in \mathbb{R}^d$ parameterized by $\boldsymbol{\theta} \in \mathbb{R}^p$. Training starts from an initial state $\boldsymbol{\xi}_0 = (\boldsymbol{\theta}_0, \boldsymbol{v}_0)$, where $\boldsymbol{v}$ is the velocity for momentum-based optimizers, typically the zero vector when initialized. Optimization proceeds for $K$ steps using a dataset $\mathcal{D}$, typically partitioned into ordered batches $\mathcal{B}_0, \mathcal{B}_1, \ldots, \mathcal{B}_{K-1}$. At each step $k$ (from $0$ to $K-1$), the parameters and velocity are updated by gradients $G_k$ computed on a data batch $\mathcal{B}_k$ and a loss function $\mathcal{L}$ through optimizers with a learning rate $\eta_k$, a momentum factor $\mu$, and a weight decay factor $\lambda$. This sequence of gradient-based updates defines the observed state trajectory $\boldsymbol{\xi}_k = (\boldsymbol{\theta}_k, \boldsymbol{v}_k)$ for $k = 0, \ldots, K$ including the parameters $\boldsymbol{\theta}$ and velocity $\boldsymbol{v}$. The specific update rule considered in this paper is SGD with momentum and weight decay (Sutskever et al., 2013), with the implementation closely following modern deep learning frameworks like PyTorch (Paszke et al., 2017):

$$G_k = \frac{1}{|\mathcal{B}_k|} \sum_{\boldsymbol{x} \in \mathcal{B}_k} \nabla \mathcal{L}(\boldsymbol{\theta}_k, \boldsymbol{x}) \tag{1}$$

$$G_k{}^{wd} = G_k + \lambda \boldsymbol{\theta}_k \tag{2}$$

$$\boldsymbol{v}_{k+1} = \mu \boldsymbol{v}_k + G_k{}^{wd} \tag{3}$$

$$\boldsymbol{\theta}_{k+1} = \boldsymbol{\theta}_k - \eta_k \boldsymbol{v}_{k+1} \tag{4}$$

### 3.2. Causal Analysis with Potential Outcomes

To formally analyze accountability, we utilize the **potential outcomes** framework (Rubin, 1974; 2005), which provides a rigorous foundation for describing the causal effect of an intervention (**treatment**) on a target quantity (**outcome**).

Let $T \in \{0, 1\}$ denote a binary treatment assignment variable, representing the intervention to be studied, e.g., whether a training stage has occurred during model training ($T = 1$) or not ($T = 0$). The outcome variable $Y$ represents our quantity of interest affected by the treatment, such as the model's final performance. To properly define the causal effect of $T$ on $Y$, we must consider two scenarios: the outcome when $T = 0$ and when $T = 1$. For the final model, only one of these scenarios will be observed, and the other will be counterfactual. The potential outcomes framework provides the formal notation to represent both scenarios.

**Definition 3.1.** *The **potential outcome** $Y(T)$ represents the value that the outcome variable $Y$ would attain if the treatment assignment were set to $T$.*

**Definition 3.2.** *The **causal effect** $\tau$ of the treatment, also known as the individual treatment effect (ITE), is defined as the difference between the potential outcomes under treatment and control: $\tau = Y(1) - Y(0)$.*

A fundamental challenge in causal inference is that we can only observe one outcome $Y$—normally the one corresponding to the treatment actually received (Holland, 1986). The **consistency property** (Cole & Frangakis, 2009) establishes the relationship between this observed outcome $Y$ and the potential outcome under the received treatment $Y(T)$, i.e., $Y(1) = Y$. To estimate $\tau$, we must develop methods to estimate the unobserved counterfactual outcome $Y(0)$.

## 4. A Framework for Accountability Attribution

### 4.1. Problem Formalization: Causal Effect of a Stage

We define the accountability attribution problem as the causal effect of a model development stage on the final model behavior. For each stage, this problem corresponds to the counterfactual question: how would the model's behavior have changed if the updates from the stage had not occurred? We present our framework for the accountability attribution problem building upon the general causal framework in §3.2, by specifying the treatment as whether the stage has occurred or not. Formally, for a development process with model parameters evolving from $\boldsymbol{\theta}_0$ to $\boldsymbol{\theta}_K$, let $S = \{t_1, \ldots, t_s\}$ be a stage involving update steps $t_i \in \{0, \ldots, K-1\}$ for all $i \in \{1, \ldots, s\}$. The treatment $T_S \in \{0, 1\}$ indicates whether the model updates at steps in $S$ have occurred ($T_S = 1$) or all steps in $S$ have not occurred ($T_S = 0$). At each time step $k$, we define

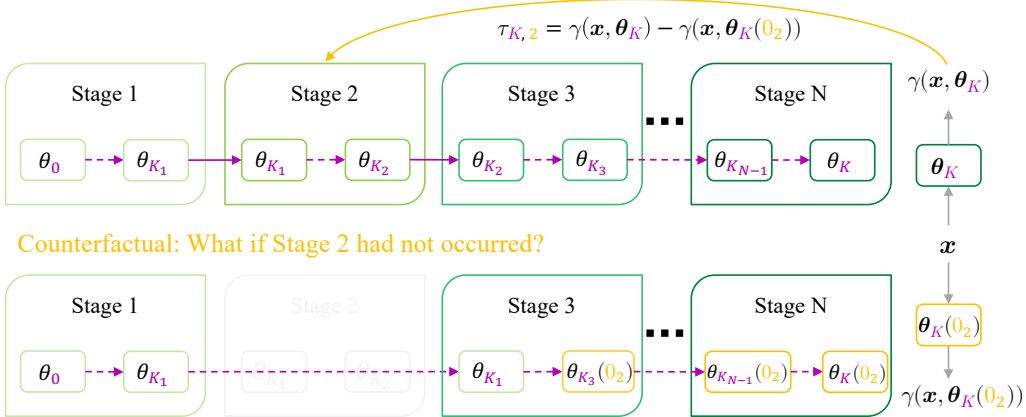

*Figure 2.* Illustration of the accountability attribution problem for a stage. A model is developed in $N$ stages, each comprising a sequence of parameter updates. The goal is to estimate the causal effect of a stage, e.g. Stage 2, on the final model behavior. The diagram shows the actual development process (top) and a counterfactual process (bottom) where Stage 2 had not occurred. The accountability of Stage 2 for predicting an input $x$ is quantified by the performance difference of the observed ($\theta_K$) and counterfactual ($\theta_K(0_2)$) model.

the potential outcome of the state under the treatment as $\boldsymbol{\xi}_k(T_S) = (\boldsymbol{\theta}_k(T_S), \boldsymbol{v}_k(T_S))$.

- The observed (treated) state trajectory corresponds to $T_S = 1$, which we denote as $\boldsymbol{\xi}_k(1_S)$. By the consistency property, $\boldsymbol{\xi}_k(1_S) = \boldsymbol{\xi}_k$.

- The counterfactual (controlled) state trajectory corresponds to $T_S = 0$, which we denote as $\boldsymbol{\xi}_k(0_S)$. It evolves by executing the standard update for $k \notin S$ and skipping the update for $k \in S$. That is, if $k \in S$, then $\boldsymbol{\xi}_{k+1}(0_S) = \boldsymbol{\xi}_k(0_S)$.

We use a performance function $\gamma(\boldsymbol{x}, \boldsymbol{\theta})$ to quantify the model's behavior on an instance $\boldsymbol{x}$ at a given state with parameters $\boldsymbol{\theta}$, for example, the performance can be the log-likelihood $\log p(\boldsymbol{x}; \boldsymbol{\theta})$. For a time step $k$, we define the outcome under treatment $T_S$ as $Y_k(T_S) = \gamma(\boldsymbol{x}, \boldsymbol{\theta}_k(T_S))$.

Putting these together, the accountability attributed to the stage $S$ is formalized as the causal effect $\tau_{K,S}$ of the treatment $T_S$ at the final time $K$ on the performance $\gamma$:

$$
\begin{aligned}
\tau_{K,S} &= Y_K(1_S) - Y_K(0_S) \\
&= \gamma(\boldsymbol{x}, \boldsymbol{\theta}_K(1_S)) - \gamma(\boldsymbol{x}, \boldsymbol{\theta}_K(0_S)) \\
&= \gamma(\boldsymbol{x}, \boldsymbol{\theta}_K) - \gamma(\boldsymbol{x}, \boldsymbol{\theta}_K(0_S)).
\end{aligned} \tag{5}
$$

To solve the accountability attribution problem, any estimator for the causal effect $\tau_{K,S}$ can be plugged in to our framework. In the following section, we present our AA-Score estimator derived through state trajectory interpolation and Taylor expansion.

### 4.2. Estimating the Causal Effect of a Stage

The proposed AA-Score is an estimator $\hat{\tau}_{K,S}$ for the causal effect $\tau_{K,S}$. To derive this estimator, we first consider a

special case of the treatment only including a single step $t$, i.e., $S = \{t\}$. The estimand for this special case is noted as $\tau_{K,t}$. We write $T_S$ as $T$ ($1_S$ as $1$ and $0_S$ as $0$) for simplicity to refer to the treatment at step $t$.

To derive our estimator $\hat{\tau}_{K,t}$ for $\tau_{K,t}$, we first show an intermediate result of estimating the causal effect of the updating step $t$ on the state $\boldsymbol{\xi}$ (different from the effect on the performance $\gamma$) at any time $k$. We denote this effect on the state as $\boldsymbol{w}_{k,t} = \boldsymbol{\xi}_k(1) - \boldsymbol{\xi}_k(0)$ and derive our estimator $\hat{\boldsymbol{w}}_{k,t}$ by propagating the initial effect at step $k = t + 1$.

**Estimator 4.1** (Single step effect on a state). *Let $\boldsymbol{w}_{k,t}$ be the causal effect of the updating step $t$ on the state $\boldsymbol{\xi}_k$. For step $k = t + 1$, the estimator $\hat{\boldsymbol{w}}_{t+1,t}$ computes the exact effect $\boldsymbol{w}_{k,t}$ as the state difference:*

$$
\boldsymbol{w}_{t+1,t} = \hat{\boldsymbol{w}}_{t+1,t} = \begin{pmatrix} \boldsymbol{\theta}_{t+1} - \boldsymbol{\theta}_t \\ \boldsymbol{v}_{t+1} - \boldsymbol{v}_t \end{pmatrix} = \begin{pmatrix} -\eta_t \boldsymbol{v}_{t+1} \\ \boldsymbol{v}_{t+1} - \boldsymbol{v}_t \end{pmatrix} \tag{6}
$$

*For all steps $k \in \{t+1, \ldots, K-1\}$, define the one-step propagator matrix as:*

$$
\mathbf{M}_k = \begin{pmatrix} \mathbf{I} - \eta_k(H_k + \lambda\mathbf{I}) & -\eta_k \mu\mathbf{I} \\ H_k + \lambda\mathbf{I} & \mu\mathbf{I} \end{pmatrix} \tag{7}
$$

*where $H_k = \sum_{\boldsymbol{x} \in \mathcal{B}_k} \nabla^2 \mathcal{L}(\boldsymbol{\theta}_k, \boldsymbol{x})$ is the Hessian of the loss $\mathcal{L}$ evaluated at the observed $\boldsymbol{\theta}_k$. The matrix $\mathbf{M}_k$ propagates the estimated effect on the state from step $k$ to step $k + 1$ as $\hat{\boldsymbol{w}}_{k+1,t} = \mathbf{M}_k \hat{\boldsymbol{w}}_{k,t}$. Define the overall propagator matrix to a target time $k$ as:*

$$
\mathbf{P}^{((t+1) \to k)} = \begin{cases} \prod_{i=k-1}^{t+1} \mathbf{M}_i & \text{if } t + 1 < k \leq K \\ \mathbf{I} & \text{if } k = t + 1 \end{cases}
$$

*Then, the estimator $\hat{\boldsymbol{w}}_{k,t}$ can be computed by propagating the initial effect:*

$$
\hat{\boldsymbol{w}}_{k,t} = \mathbf{P}^{((t+1) \to k)} \hat{\boldsymbol{w}}_{t+1,t}. \tag{8}
$$

By plugging in the performance function $\gamma$ into the state-effect estimator $\hat{\boldsymbol{w}}_{K,t}$, we can derive our estimator $\hat{\tau}_{K,t}$ for the effect on the final model behavior.

**Estimator 4.2** (Single step effect on the model behavior). *Let $\tau_{K,t}$ be the causal effect of the updating step $t$ at the final time $K$ on the model behavior quantified by the performance function $\gamma$. Let $\mathbf{P}^{((t+1)\rightarrow K)} = \begin{pmatrix} \mathbf{P}_{11} & \mathbf{P}_{12} \\ \mathbf{P}_{21} & \mathbf{P}_{22} \end{pmatrix}$ be the overall propagator matrix from $t+1$ to $K$ as defined in Estimator 4.1. Let $E_t = \mathbf{P}_{11}(-\eta_t \boldsymbol{v}_{t+1}) + \mathbf{P}_{12}(\boldsymbol{v}_{t+1} - \boldsymbol{v}_t)$ be the estimated effect on the parameters at time $K$ (first block of the estimated effect on the state vector $\hat{\boldsymbol{w}}_{K,t}$). The estimator for $\tau_{K,t}$ can be computed as a dot product:*

$$\hat{\tau}_{K,t} = \nabla_{\boldsymbol{\theta}}\gamma(\boldsymbol{x}, \boldsymbol{\theta}_K)^\top E_t, \qquad (9)$$

*where the gradient $\nabla_{\boldsymbol{\theta}}\gamma$ is evaluated at the observed final parameters $\boldsymbol{\theta}_K$.*

The detailed derivation of Estimators 4.1 and 4.2 is deferred to App. A.1. Importantly, we see that eq. (9) is the product of two parts. The second part $E_t$ depends on the model development process and the treatment step $t$ but is independent of the test instance $\boldsymbol{x}$ or the performance function $\gamma$. This allows $E_t$ to be computed only once along the model development and reused to efficiently estimate the performance effect on any new data instance or a new set of data instances by doing a dot product. The computational considerations and efficient implementation strategies for computing $E_t$ are discussed in §4.4.

Once we establish the estimators $\hat{\boldsymbol{w}}_{K,t}$ and $\hat{\tau}_{K,t}$ for the causal effect of a single step, we can extend them to estimate the effect of an entire stage. We show in the following that the total effect of a stage can be estimated as the sum of the individual effects for each step in the stage.

**Estimator 4.3** (Effect of a stage). *Let $S = \{t_1, \ldots, t_s\}$ be a stage with steps $t_i \in \{0, \ldots, K-1\}$ for all $i \in \{1, \ldots, s\}$. Let $\boldsymbol{w}_{K,S}$ and $\tau_{K,S}$ denote the causal effects of stage $S$ at the final time $K$ on the state and the model behavior, respectively. Let $\hat{\boldsymbol{w}}_{K,t_i}$ and $\hat{\tau}_{K,t_i}$ be the single-step effect estimators of step $t_i$, as defined in Estimators 4.1 and 4.2. The first-order estimators $\hat{\boldsymbol{w}}_{K,S}$ and $\hat{\tau}_{K,S}$ can be computed as the sum of the single-step effect estimators for each step in $S$:*

$$\hat{\boldsymbol{w}}_{K,S} = \sum_{t_i \in S} \hat{\boldsymbol{w}}_{K,t_i}, \quad \hat{\tau}_{K,S} = \sum_{t_i \in S} \hat{\tau}_{K,t_i} \qquad (10)$$

The proof, based on the linearity of the first-order approximation derived from a multi-parameter Taylor expansion, is provided in App. A.2. The estimated effect $\hat{\tau}_{K,S}$ in eq. (10) will be the AA-Score of the training stage $S$.

### 4.3. Accuracy of the Estimators

The estimators derived above rely on Taylor approximations, which raises a natural question: how accurate are these estimators? The approximation error depends on how much the counterfactual trajectory $\boldsymbol{\xi}_k(0)$ deviates from the observed trajectory $\boldsymbol{\xi}_k(1)$ as they evolve through the model development process. When the trajectories remain close, the approximation accurately captures the causal effect. However, as the trajectories diverge, higher-order terms in the Taylor expansion become significant, leading to larger approximation errors. We characterize this trade-off by deriving an upper bound on the estimation error that depends on properties of the loss landscape, the model development dynamics, and the magnitude of the initial effects on the state.

**Theorem 4.1** (Error Bound for Stage Effect Estimation). *Assume the loss function $\mathcal{L}$ is twice differentiable with an L-Lipschitz continuous Hessian, and the optimization update maps are locally stable on the region between the observed and counterfactual trajectories, e.g., for the state update map $\Psi_k$, $\|D\Psi_k(\boldsymbol{\xi}_k)\|_2 \leq e^{\eta_k\Lambda}$. Let $\eta \geq \eta_k$ be the maximum learning rate. Let $D_S := \sum_{t_i \in S} \|\boldsymbol{w}_{t_i+1,t_i}\|$ be the magnitude of the initial effects on the state by stage $S$. Let the first and second derivatives of the performance function $\gamma$ be bounded. Then, the estimation error of the effect of stage $S$ on the final model performance is bounded by:*

$$|\tau_{K,S} - \hat{\tau}_{K,S}| \leq \frac{L_\gamma D_S^2}{2} e^{2\eta\Lambda(K-\min(S))} \qquad (11)$$

We prove the error bound in App. B. The bound scales linearly with the effective constant $L_\gamma$, which combines the Lipschitz constant $L$ and the derivatives of the performance function $\gamma$. It scales with $D_S^2$, which represents the magnitude of the initial differences of the trajectories. The error grows exponentially with the learning rate $\eta$, the instability constant $\Lambda$, and the remaining time horizon $K-\min(S)$. Attribution is most reliable for **late** stages in the development process (small $K - \min(S)$) or in **stable** regimes (small $\Lambda$). Early-stage effects in chaotic regimes are fundamentally hard to attribute because changes accumulate to divergence.

### 4.4. Computational Considerations

To estimate the effect of a training stage, we need to compute $E_t$ for each step in the stage as outlined in Estimators 4.1 and 4.2. Although we only need to do this computation once along the training process, the direct computation of $E_t$ presents significant computational challenges. These primarily arise from the manipulation of the propagator matrices $\mathbf{M}_k$ and $\mathbf{P}^{((t+1)\rightarrow K)}$, which have size $2p \times 2p$, and the Hessian computation which is $O(p^2)$, where $p$ is the dimension of the parameter vector $\boldsymbol{\theta}$. With batch size $|\mathcal{B}|$, the complexity of the propagation will be $O(K|\mathcal{B}|p^2)$, which we discuss in detail in App. C.1.

# 5. Experiments

## 5.1. Datasets and Experiment Settings

We show our AA-Score for causal effect estimation on various datasets and tasks. MNIST (LeCun, 1998): A benchmark dataset of handwritten digit images. CELEBA (Liu et al., 2015): A large-scale dataset of facial images known to contain spurious correlations, e.g., gender with hair color. CIVILCOMMENTS (Borkan et al., 2019): A dataset of public comments labeled for toxicity and whether they contain words corresponding to demographic information such as race, gender, and religion, often used for studying fairness and bias. Lastly, a group of three chest X-ray datasets: CHESTX-RAY14 (Wang et al., 2017), PADCHEST (Bustos et al., 2020), and VINDR-CXR (Nguyen et al., 2020). They contain pathologist-annotated chest radiographs collected from patients in the U.S., Spain, and Vietnam, respectively.

For each dataset, we use model architectures appropriate to the task. We start with Multi-Layer Perceptrons (MLPs) on MNIST to facilitate detailed analysis and demonstrate the utility of our method. For CELEBA, we use ResNets (He et al., 2016) to assess our method on more complex image recognition tasks. For the CIVILCOMMENTS dataset, we fine-tune a pre-trained Transformer model, specifically a Gemma-3 (Team et al., 2025) from Huggingface (Wolf et al., 2019), to evaluate the effect of each stage in the context of fine-tuning language models. Finally, we use DenseNet-121 (Huang et al., 2017) and the TorchXRayVision (Cohen et al., 2020) framework for X-ray datasets to evaluate our method for handling data distribution shift.

We use the log-likelihood as the performance function $\gamma$ and compute the AA-Score estimator $\hat{\tau}_{K,t}$ for each stage. Therefore, a **positive** $\hat{\tau}_{K,t}$ indicates that the training stage contributes to a **higher log-likelihood**, i.e., the stage is **beneficial** to the model's performance.

Beyond these core settings, we adopt additional settings to study specific properties of AA-Score. To benchmark the practical runtime and memory overhead on CELEBA with ResNet and CIVILCOMMENTS with Pythia (Biderman et al., 2023), varying both dataset size and model size. To assess robustness in an adversarial setting, we disperse Bad-Nets (Gu et al., 2019) poisoned data through CIFAR-10 training on ResNet18 following BackdoorBench (Wu et al., 2022). We conduct all experiments on a cluster using one NVIDIA A100 GPU with 40G memory. More details of experiment settings are in App. D.1.

## 5.2. Accountability Attribution Demonstration

The first set of results is on MNIST. We consider several semi-synthetic settings to demonstrate the utility of AA-Score. Main results are in Fig. 3, where we show the effects of all update steps. Additional results are in App. E.

**Capture an influential positive stage** One application of accountability attribution is to identify stages that make significant contributions to the model's performance. These stages provide useful learning signals for the model development and are responsible for the model's success. To show AA-Score can identify such stages, we consider a simple setting with one stage consisting of one update step. Specifically, we exclude all instances of a digit (e.g., digit '4') from the training set. Then, at a specific step, we insert a data point of the digit '4' (a stage with one update step), which will be the only stage providing the learning signal for recognizing '4' to the model. We estimate the effect of all steps, and show that the inserted step has a high positive effect on the model's performance on classifying '4' (tested on the same image and other '4' images), demonstrating AA-Score can identify stages processing influential updates to the model. Results are in Fig. 3 (a).

**Capture a negative stage caused by mislabeled data** Besides positive stages, it is critical to detect stages that negatively impact the model's performance for assigning blame. Since a main reason for negative effects is data quality issues, we demonstrate that AA-Score can capture negative effects of stages by creating mislabeled data. In particular, we conduct the standard training procedure but modify the labels of a small subset of training data. We show that the stage with steps processing mislabeled data has a negative effect on the final model's performance according to AA-Score, demonstrating AA-Score can capture negative effects of stages. Results are shown in Fig. 3 (b).

**Attribute to stages with data distributional shifts** For multi-stage development, data distributional shifts can occur naturally in continual learning or domain adaptation. Understanding how a stage with a different data distribution affects the final model is crucial for diagnosing issues like catastrophic forgetting or identifying spurious correlations. To show the effect of AA-Score in cases with distributional shifts, we perform a 3-stage development on MNIST with transformed data. Stage 1: standard MNIST. Stage 2: MNIST images rotated by 45 degrees. Stage 3: MNIST images rotated by 90 degrees. We then evaluate the effect of each stage on the model's performance on test sets from each distribution (original, 45-degree rotated, and 90-degree rotated). A stage consisting of training data from the same distribution as the test data is called the *in-distribution* (ID) stage, and the stage is expected to have a high positive effect on the model's performance. A stage consisting of training data from a different distribution from the test data is called the *out-of-distribution* (OOD) stage, and the model will have to adapt to the new distribution and is expected to have a small or even negative effect on the model's performance. We observe that the effect estimated by AA-Score follows the expected pattern, shown in Fig. 3 (c-e).

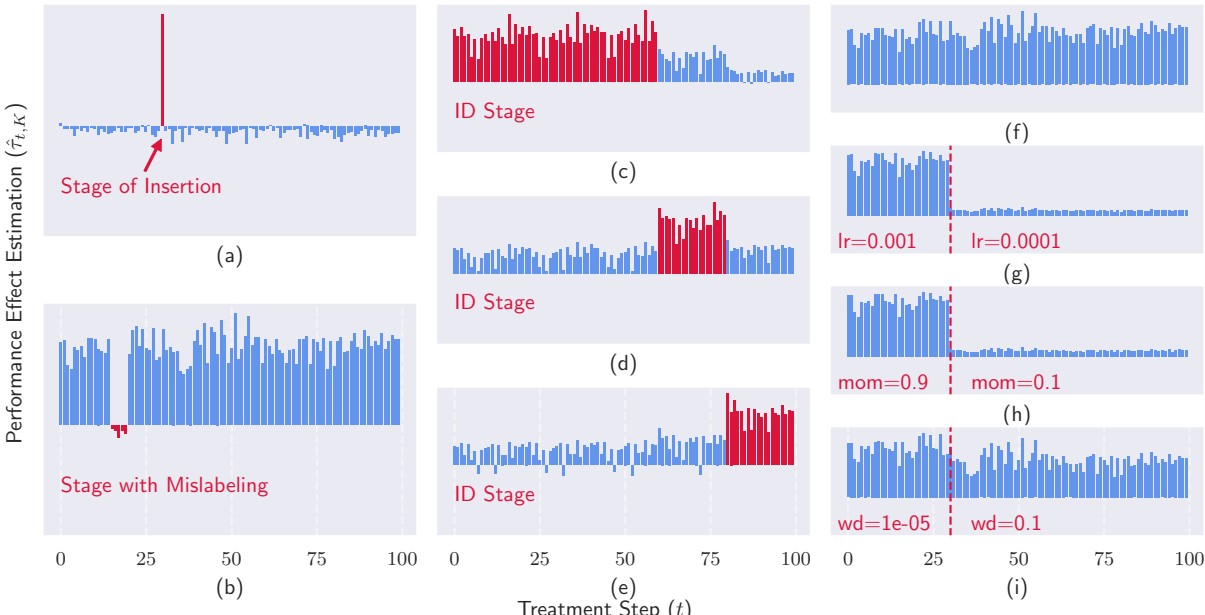

*Figure 3.* Performance effect on MNIST. Each bar shows the AA-Score estimation ($\hat{\tau}_{K,t}$) for an update step $t$, which can be aggregated to stage effects. A **positive** $\hat{\tau}_{K,t}$ indicates that the stage leads to a **higher log-likelihood**, i.e., the stage is **beneficial**. (a) Accurately detect an influential stage of an inserted data point. (b) Capture a stage processing mislabeled data, demonstrating their negative effect on the test performance. (c-e) The stage with the highest effect on the test set is the in-distribution (ID) training stage. (c) original test set. (d) 45-degree rotated test set. (e) 90-degree rotated test set. (f) Baseline for optimization parameters. (g) Higher/lower lr leads to higher/lower performance effect. (h) Higher/lower mom leads to higher/lower performance effect. (i) Higher/lower wd leads to slightly lower/higher performance effect.

**Reflect the influence of optimization parameters** An important difference between AA-Score and other data-centric attribution methods is that AA-Score can capture not only the influence of data but also the influence of optimization parameters for effect estimation. This allows AA-Score to drop assumptions about the model's development process (e.g., convexity, convergence, permutation invariance) and assign accountability for the specific model instead of the "average model" that is expected to be developed from the data. We demonstrate this by showing the AA-Score estimation when important optimization parameters are varied, including the learning rate (lr), momentum (mom), and weight decay (wd). We vary these parameters to separate the development into two stages with different optimization parameters, and analyze the stage effects and observe their influence. When we vary each parameter, we keep the other parameters the same across the two stages. We show the results in Fig. 3 (f-i). (f) is the baseline with only one stage, where all three parameters stay constant: lr=0.001, mom=0.9, and wd=1e-5. These are common values for training MLPs on MNIST. For experiments in (g-i), we use these common values for the stage 1 and begin a stage 2 with different parameter values at training step 30. In (g), we set the lr to 0.0001 (decreased by 10x) in stage 2. In (h), we set the mom to 0.1 (decreased by 9x) in stage 2. In

(i), we set the wd to 0.1 (increased by 10000x) in stage 2. We see that as the lr decreases, AA-Score decreases as well. This is expected because the second stage with smaller lr has less impact on the model's parameters. We also see that as mom decreases, AA-Score decreases because smaller mom leads to less impact on the results. Finally, as wd increases, stage 2's AA-Score becomes smaller because the meaningful learning signals from the data are less significant with larger wd. The changes are not as significant as the lr and mom changes because wd is not a direct factor for updating the model's parameters but a regularization term mixed with the gradients (eq. (1)). These results work as sanity checks, as they show that our method can capture the effect of optimization parameters on model performance.

**Evaluate against retraining** For the cases of insertion, mislabeled data, and distributional shifts, because we know the specific time steps that the stage contains, we also retrain the model to get the counterfactual effect of skipping that stage as the ground-truth for evaluation, e.g., model performance when no inserted or mislabeled stage, or skipping one of the shifted stages. For insertion and mislabeled batches, we skip only the corresponding update steps inside the original training loop, while the learning-rate scheduler advances normally and momentum is not updated for skipped steps because their gradients are not observed. For

*Table 1.* Correlation on MNIST between estimated stage effects and counterfactual results obtained by retraining with the stage skipped. Results are computed over model performance on a test set of randomly sampled data points. High correlation indicates that the estimator successfully captures the effect of the stage.

| Method | Insert | Mislabel | Shift 1 | Shift 2 | Shift 3 | Avg |
|---|---|---|---|---|---|---|
| DVEmb | 0.9441 | 0.9496 | 0.9521 | 0.9329 | 0.9297 | 0.9417 |
| AA-Score | 0.9444 | 0.9487 | 0.9518 | 0.9350 | 0.9482 | 0.9456 |

full distribution-shift stages, the counterfactual retraining follows the same multi-stage pipeline in practice except that the target stage is skipped. Subsequent stages start from the checkpoint before the skipped stage with their optimizer state and learning-rate scheduler reinitialized. We then compare the estimated effects with the ground-truth counterfactual results. For comparison, we also add an adapted version of DVEmb (Wang et al., 2025) as a baseline, originally a process-based data attribution method, which we have adapted to the stage-level setting by aggregating data-point effects within each batch and then across batches in a stage. In Table 1, we show the correlation between the estimated effects and the ground-truth results on a test set of randomly sampled MNIST data points. Both DVEmb and AA-Score achieve high correlation with retraining results on MNIST, while AA-Score has a slightly higher average correlation 0.9456. This high correlation indicates that AA-Score can accurately capture the effect of training stages on model performance quantitatively.

### 5.3. Accountability Attribution for Image and Text Classification

We then show more practical applications of AA-Score for ResNet on CELEBA and X-ray datasets and a fine-tuned Gemma-3 on CIVILCOMMENTS.

**Detect spurious correlations** We investigate whether our method can identify spurious correlations, i.e., features that are predictive during model development but not causally related to the target label, and guide targeted follow-up interventions. We examine two benchmark datasets with documented spurious features: CELEBA and CIVILCOMMENTS. In CELEBA, we study the binary classification task of predicting whether a person is *blonde*, where hair color is spuriously correlated with *gender* (Koh et al., 2021). In CIVILCOMMENTS, we analyze toxicity detection, where the *demographic identity terms* like race, gender, and religion in the comments are spuriously correlated with toxicity (Borkan et al., 2019).

For each dataset, we designate the ground-truth label (blonde or toxicity) as the *real label* and the spuriously correlated feature (gender or demographic identity) as the *confounding feature*. We hypothesize that spurious correlations emerge during specific stages, and removing these

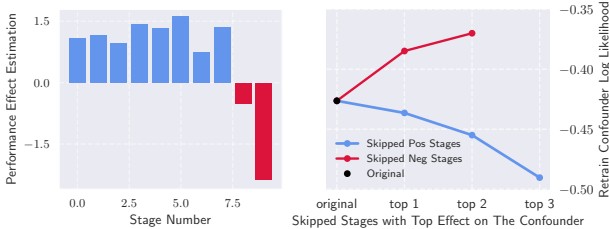

*Figure 4.* Performance effect estimation (left) and retraining likelihood with confounding features as the label (right) on CIVILCOMMENTS. There are 8 positive and 2 negative stages. We show the likelihood decreases when skipping the top 3 positive stages and increases when skipping the 2 negative stages.

stages reduces the model's reliance on confounding features. Therefore, we first conduct standard model training, separate all the training steps into 10 stages, and compute AA-Score for each stage on model performance. Then, we select the top stages with strongest positive or negative effects on predicting the confounding feature, and retrain the model while skipping the selected stages. Our results in Fig. 4 confirm the hypothesis on CIVILCOMMENTS. For CELEBA, we get similar results where the confounder log likelihood decreases by 0.1130 when skipping the top two stages and increases by 0.0856 when skipping the bottom stage. Skipping stages with strong positive effects on the confounding feature reduces the model's likelihood on the confounding label while skipping stages with negative effects increases the likelihood. These findings demonstrate that AA-Score can localize origins of spurious correlations and provide actionable interventions.

**Evaluate against retraining** We further conduct 3-stage development on CELEBA, CIVILCOMMENTS, and X-ray datasets with different data distributions and compute AA-Score for each stage on model performance. We retrain the model to get the counterfactual effect of skipping that stage as the ground-truth for evaluation, using the same multi-stage-skipping protocol as in the MNIST data distributional shift experiment. We then compare the estimated effects with the ground-truth results and the stage-level DVEmb baseline. In Table 2, AA-Score achieves high correlation with retraining results and outperforms DVEmb in most complex multi-stage settings. This improvement is consistent with our design. AA-Score directly targets stage-level counterfactual effects and accounts for complex optimization dynamics, whereas DVEmb was originally developed for data-instance-level attribution and assumes simpler SGD dynamics.

### 5.4. Runtime and Memory

We further evaluate the practical overhead in terms of runtime and memory for computing AA-Score. In general, the computational cost has two parts: (1) *logging*, which

*Table 2.* Correlation on CELEBA, CIVILCOMMENTS, and X-ray datasets between estimated stage effects and counterfactual results obtained by retraining with the stage skipped. Results are computed over model performance on a randomly sampled test set. High correlation indicates that the estimator successfully captures the effect of the stage.

| Dataset | Method | Stage 1 | Stage 2 | Stage 3 |
|---|---|---|---|---|
| CelebA | DVEmb | 0.4406 | 0.6304 | 0.4203 |
| | AA-Score | 0.5524 | 0.6953 | 0.8436 |
| CivilComments | DVEmb | 0.7766 | 0.9569 | 0.8514 |
| | AA-Score | 0.8576 | 0.9789 | 0.9746 |
| X-Ray | DVEmb | 0.9628 | 0.9487 | 0.7025 |
| | AA-Score | 0.9326 | 0.9809 | 0.7414 |

occurs during training and saves gradients and optimizer states, and (2) *attribution*, which is run after training to propagate perturbations. On CELEBA with ResNet and CIVILCOMMENTS with Pythia, we find that attribution time is manageable for the small to medium-sized settings and scales roughly linearly with both dataset size and model size. For runtime, logging can be more expensive when I/O dominates training (full training), and its relative overhead is much lower when backpropagation dominates I/O (e.g., LoRA fine-tuning). For memory, peak memory is typically determined either by training or by the largest layer-wise attribution computation, and it is feasible unless there is one very large layer. Detailed runtime, memory, and scaling-correlation results are reported in App. C.2.

### 5.5. Robustness to Dispersed Backdoor Attacks

We further test the robustness of AA-Score to an adversarial setting where BadNets poisoned data are dispersed through CIFAR-10 training on ResNet18 following BackdoorBench. This setting is challenging because the poisoned examples are not isolated in any stages but are scattered across the whole process. We consider two settings: one where poisoned examples are concentrated in a few batches and another where they are spread evenly across all batches. The results show that AA-Score can still reveal poisoned batches but with less discriminative power for the fully dispersed setting. This behavior is expected, since attribution operates at the batch level and mixed batches offer little contrast for distinguishing poisoned from regular examples. These findings suggest that AA-Score remains useful for accountability attribution under realistic attacks, while highlighting the limits when adversaries deliberately disperse the poisoning signals. Details are in App. E.6.

## 6. Discussion and Limitations

In our framework, the choice of stage determines the granularity of the accountability question rather than adding a separate modeling assumption. In many development pipelines, stages are naturally available, such as pretraining, fine-tuning, alignment, and different data sources. When such prior structure is unavailable, the same formalism can treat each data batch or update step as a candidate stage and then aggregate step-level scores into stage-level scores depending on the accountability question. Changing the granularity affects the interpretation of the scores, but the attribution procedure still computes reusable scores along the observed trajectory. Moreover, the method should be viewed as an auditing and diagnostic tool rather than an automatic remedy. High-magnitude positive or negative scores identify stages worth inspecting, but responsible mitigation requires additional steps considering the deployment context and possible side effects of intervention.

Our work has several limitations that suggest future research directions. First, our current estimators rely on first-order approximation, which may be insufficient when higher-order effects are significant. Future work could incorporate higher-order approximations or learned propagator surrogates to improve accuracy. Second, despite being more efficient than retraining, computational cost can be high for large-scale models due to high-dimensional propagator and Hessian matrix computations, as discussed in §4.4. Structured approximations (layer-wise, block-wise, and parameter-subset) and distributed computing could help scale the framework to foundation models. Its applicability to large-scale models remains an open research question. Finally, our framework assumes that the trajectory, optimizer states, and hyperparameters are recorded. When such information is inaccessible, applying the method may require additional approximations, such as reconstructing the trajectory from major checkpoints.

## 7. Conclusion

In this paper, we introduced the problem of accountability attribution, which traces model behavior to specific stages of the training process. Our key contributions include: formulating this novel accountability attribution problem; developing a general framework based on potential outcomes and counterfactual queries about training stage effects; deriving efficient estimators that account for complex optimization dynamics like learning rate schedules, momentum, and weight decay; and demonstrating practical utility by uncovering influential stages responsible for beneficial or harmful model behaviors across diverse settings. Empirically, we showed how our framework enables attributing model behavior to training stages in a principled way. This work takes a step toward more transparent, interpretable, and accountable AI development by providing tools to analyze and assign responsibility within complex training pipelines.

## Impact Statement

The goal of this work is to make the development of AI systems more transparent by attributing model behavior to specific stages. Potential positive impacts include helping developers and auditors diagnose harmful model behavior, identify stages that potentially introduced spurious correlations or poisoning signals, as well as allocate credit for beneficial development choices. These benefits may support more accountable model documentation and targeted investigation before deployment. The same attribution results can potentially be misused or misinterpreted. They should not be treated as definitive evidence of individual fault, especially in collaborative settings where a stage may reflect many upstream choices about data collection, labeling, infrastructure, and deployment constraints. Moreover, malicious actors could also use auditing feedback to hide problematic training signals or selectively disclose only favorable results. We therefore recommend using accountability attribution as one component of a broader audit practice.

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

# A. Derivation of Estimators 4.1 to 4.3

Here we provide the detailed steps to derive the results stated in Estimators 4.1 to 4.3.

## A.1. Estimators 4.1 and 4.2

We first consider the special case of treatment on a single step $t$ as in Estimators 4.1 and 4.2. Recall that the treatment variable $T \in \{0, 1\}$ is defined such that $T = 0$ means step $t$ is skipped (counterfactual), and $T = 1$ means step $t$ is executed (observed). The causal effect of $T$ on the state at any time $k$ is $\boldsymbol{w}_{k,t} = \boldsymbol{\xi}_k(1) - \boldsymbol{\xi}_k(0)$. For the model behavior, we consider the effect at the final time $K$ with performance function $\gamma(\boldsymbol{x}, \boldsymbol{\theta})$, which is $\tau_{K,t} = \gamma(\boldsymbol{x}, \boldsymbol{\theta}_K(1)) - \gamma(\boldsymbol{x}, \boldsymbol{\theta}_K(0))$.

We introduce an interpolation parameter $\epsilon \in [0, 1]$ that defines a continuous path between the observed state $\boldsymbol{\xi}_k(1)$ at $\epsilon = 1$ and the counterfactual state $\boldsymbol{\xi}_k(0)$ at $\epsilon = 0$, with the interpolated states $\boldsymbol{\xi}_k(\epsilon) = \boldsymbol{\xi}_k(0) + \epsilon(\boldsymbol{\xi}_k(1) - \boldsymbol{\xi}_k(0))$.

For step $k$ up to $t$, $\boldsymbol{\xi}_k(\epsilon) = \boldsymbol{\xi}_k$ because the observed and counterfactual states are the same. At step $t + 1$, whether the update step $t$ had occurred or not causes a difference of the two states. The interpolated state $\boldsymbol{\xi}_k(\epsilon)$ thus becomes:

$$\boldsymbol{\theta}_{t+1}(\epsilon) = \boldsymbol{\theta}_t + \epsilon(\boldsymbol{\theta}_{t+1} - \boldsymbol{\theta}_t) = \boldsymbol{\theta}_t - \epsilon \eta_t \boldsymbol{v}_{t+1} \tag{12}$$

$$\boldsymbol{v}_{t+1}(\epsilon) = \boldsymbol{v}_t + \epsilon(\boldsymbol{v}_{t+1} - \boldsymbol{v}_t) \tag{13}$$

For step $k$ after $t + 1$, $\boldsymbol{\xi}_k(\epsilon)$ evolves from $\boldsymbol{\xi}_{t+1}(\epsilon)$ as the result of the divergence between the observed and counterfactual states. This construction ensures $\boldsymbol{\xi}_k(\epsilon = 0) = \boldsymbol{\xi}_k(T = 0)$ and $\boldsymbol{\xi}_k(\epsilon = 1) = \boldsymbol{\xi}_k(T = 1)$.

**Estimator for effect on the state:** For any target time $k \geq t + 1$, the first-order Taylor expansion of $\boldsymbol{\xi}_k(\epsilon)$ around $\epsilon = 1$ (the observed path) is[1]:

$$\boldsymbol{\xi}_k(\epsilon) \approx \boldsymbol{\xi}_k(1) + \frac{\partial \boldsymbol{\xi}_k(\epsilon)}{\partial \epsilon}\bigg|_{\epsilon=1} (\epsilon - 1) \tag{14}$$

Get $\boldsymbol{\xi}_k(0)$ with the approximation and plug it into the effect on the state:

$$\boldsymbol{w}_{k,t} = \boldsymbol{\xi}_k(1) - \boldsymbol{\xi}_k(0) \approx \boldsymbol{\xi}_k(1) - \left(\boldsymbol{\xi}_k(1) - \frac{\partial \boldsymbol{\xi}_k(\epsilon)}{\partial \epsilon}\bigg|_{\epsilon=1}\right) = \frac{\partial \boldsymbol{\xi}_k(\epsilon)}{\partial \epsilon}\bigg|_{\epsilon=1} \tag{15}$$

Let $\hat{\boldsymbol{w}}_{k,t} = \frac{\partial \boldsymbol{\xi}_k(\epsilon)}{\partial \epsilon}\big|_{\epsilon=1}$ denote the first-order approximation. Then $\boldsymbol{w}_{k,t} \approx \hat{\boldsymbol{w}}_{k,t}$.

**Behavior effect estimator at the final time:** For the effect on the model behavior at time $K$ with performance function $\gamma$, we have

$$\tau_{K,t} \approx \frac{\partial}{\partial \epsilon} \gamma(\boldsymbol{x}, \boldsymbol{\theta}_K(\epsilon))\bigg|_{\epsilon=1} \tag{16}$$

Then, apply the chain rule:

$$\tau_{K,t} \approx \nabla_{\boldsymbol{\theta}} \gamma(\boldsymbol{x}, \boldsymbol{\theta}_K(1))^{\top} \frac{\partial \boldsymbol{\xi}_K(\epsilon)}{\partial \epsilon}\bigg|_{\epsilon=1} = \nabla_{\boldsymbol{\theta}} \gamma(\boldsymbol{x}, \boldsymbol{\theta}_K)^{\top} [\hat{\boldsymbol{w}}_{K,t}]_{\boldsymbol{\theta}} \tag{17}$$

where $[\hat{\boldsymbol{w}}_{K,t}]_{\boldsymbol{\theta}}$ is the first block of the estimated effect on the state at the final time, i.e., the estimated effect on parameters $\boldsymbol{\theta}$ at time $K$.

**Recursive computation of effect on the state:** The estimator for the effect on the state $\hat{\boldsymbol{w}}_{k,t}$ for any target time $k \geq t + 1$ can be computed recursively from the initial effect $\boldsymbol{w}_{t+1,t}$.

**Base Case (at $k = t + 1$):** Differentiating eqs. (12) and (13) w.r.t. $\epsilon$ (the derivative is constant):

$$\frac{\partial \boldsymbol{\theta}_{t+1}(\epsilon)}{\partial \epsilon} = -\eta_t \boldsymbol{v}_{t+1}$$

$$\frac{\partial \boldsymbol{v}_{t+1}(\epsilon)}{\partial \epsilon} = \boldsymbol{v}_{t+1} - \boldsymbol{v}_t$$

---

[1]In the literature of data attribution, e.g., influence functions (Koh & Liang, 2017), similar Taylor expansions are usually used around $\epsilon = 0$. Here we use $\epsilon = 1$ because we intend to have $\epsilon = 1$ match $T = 1$. We highlight that our expansion is equivalent to the influence function expansion, as both is around the observed outcome. The difference is that the influence function defines $\epsilon = 0$ to be the observed outcome, and it is counter-intuitive to have $T = 0$ as the observed outcome in causal inference.

Thus, at the initial step, the true state difference equals the estimator:

$$\boldsymbol{w}_{t+1,t} = \hat{\boldsymbol{w}}_{t+1,t} = \begin{pmatrix} -\eta_t \boldsymbol{v}_{t+1} \\ \boldsymbol{v}_{t+1} - \boldsymbol{v}_t \end{pmatrix} \tag{6}$$

**Recursive Step (for $k > t + 1$):** Differentiating the SGD update rules (eqs. (1) to (4)) for the interpolated path w.r.t $\epsilon$ at $\epsilon = 1$:

$$\left. \frac{\partial G_k(\epsilon)}{\partial \epsilon} \right|_{\epsilon=1} = H_k \left. \frac{\partial \boldsymbol{\theta}_k(\epsilon)}{\partial \epsilon} \right|_{\epsilon=1}$$

$$\left. \frac{\partial G_k{}^{wd}(\epsilon)}{\partial \epsilon} \right|_{\epsilon=1} = (H_k + \lambda I) \left. \frac{\partial \boldsymbol{\theta}_k(\epsilon)}{\partial \epsilon} \right|_{\epsilon=1}$$

$$\left. \frac{\partial \boldsymbol{v}_{k+1}(\epsilon)}{\partial \epsilon} \right|_{\epsilon=1} = (H_k + \lambda I) \left. \frac{\partial \boldsymbol{\theta}_k(\epsilon)}{\partial \epsilon} \right|_{\epsilon=1} + \mu \left. \frac{\partial \boldsymbol{v}_k(\epsilon)}{\partial \epsilon} \right|_{\epsilon=1}$$

$$\left. \frac{\partial \boldsymbol{\theta}_{k+1}(\epsilon)}{\partial \epsilon} \right|_{\epsilon=1} = (I - \eta_k(H_k + \lambda I)) \left. \frac{\partial \boldsymbol{\theta}_k(\epsilon)}{\partial \epsilon} \right|_{\epsilon=1} - \eta_k \mu \left. \frac{\partial \boldsymbol{v}_k(\epsilon)}{\partial \epsilon} \right|_{\epsilon=1}$$

This leads to the matrix recurrence $\hat{\boldsymbol{w}}_{k+1,t} = \mathbf{M}_k \hat{\boldsymbol{w}}_{k,t}$, where

$$\mathbf{M}_k = \begin{pmatrix} \mathbf{I} - \eta_k(H_k + \lambda \mathbf{I}) & -\eta_k \mu \mathbf{I} \\ H_k + \lambda \mathbf{I} & \mu \mathbf{I} \end{pmatrix} \tag{7}$$

Unrolling the recurrence for any target time $k > t + 1$:

$$\hat{\boldsymbol{w}}_{k,t} = \left( \prod_{i=k-1}^{t+1} \mathbf{M}_i \right) \hat{\boldsymbol{w}}_{t+1,t} \tag{18}$$

Letting $\mathbf{P}^{((t+1)\to k)} = \prod_{i=k-1}^{t+1} \mathbf{M}_i$ (or $\mathbf{I}$ if $k = t + 1$), we have $\hat{\boldsymbol{w}}_{k,t} = \mathbf{P}^{((t+1)\to k)} \hat{\boldsymbol{w}}_{t+1,t}$. This establishes Estimator 4.1 for any target time $k$.

For the behavior effect at the final time $K$, we apply the estimator for the effect on the state with target time $k = K$. The estimator is $\tau_{K,t} \approx \nabla_{\boldsymbol{\theta}} \gamma(\boldsymbol{x}, \boldsymbol{\theta}_K)^\top [\hat{\boldsymbol{w}}_{K,t}]_{\boldsymbol{\theta}}$ as in eq. (17). Let $\mathbf{P}^{((t+1)\to K)} = \begin{pmatrix} \mathbf{P}_{11} & \mathbf{P}_{12} \\ \mathbf{P}_{21} & \mathbf{P}_{22} \end{pmatrix}$ be the propagator to the final time. The top block of $\hat{\boldsymbol{w}}_{K,t}$ is:

$$\begin{aligned} [\hat{\boldsymbol{w}}_{K,t}]_{\boldsymbol{\theta}} &= \mathbf{P}_{11} [\hat{\boldsymbol{w}}_{t+1,t}]_{\boldsymbol{\theta}} + \mathbf{P}_{12} [\hat{\boldsymbol{w}}_{t+1,t}]_{\boldsymbol{v}} \\ &= \mathbf{P}_{11}(-\eta_t \boldsymbol{v}_{t+1}) + \mathbf{P}_{12}(\boldsymbol{v}_{t+1} - \boldsymbol{v}_t) \\ &\overset{\text{def}}{=} E_t \end{aligned}$$

Substituting this gives the explicit form in Estimator 4.2 (eq. (9)).

### A.2. Estimator 4.3

Let $S = \{t_1, \ldots, t_s\}$ be the set of distinct steps. The treatment $T_S = 1$ means all steps in $S$ are executed, and $T_S = 0$ means all steps in $S$ are skipped. The effect on the state is $\boldsymbol{w}_{K,S} = \boldsymbol{\xi}_K(T_S = 1) - \boldsymbol{\xi}_K(T_S = 0)$.

We introduce a vector of interpolation parameters $\boldsymbol{\epsilon} = (\epsilon_{t_1}, \ldots, \epsilon_{t_s})$, where $\epsilon_{t_i} \in [0, 1]$. Let $\boldsymbol{\xi}_k(\boldsymbol{\epsilon})$ denote the state on an interpolated path. For each $t_i \in S$, $\epsilon_{t_i} = 1$ means step $t_i$ is executed, and $\epsilon_{t_i} = 0$ means step $t_i$ is skipped. For steps $k \notin S$, the standard dynamics apply (i.e., they are executed). The state where all steps in $S$ are executed is $\boldsymbol{\xi}_K(\mathbf{1})$. The state where all steps in $S$ are skipped is $\boldsymbol{\xi}_K(\mathbf{0})$.

The multivariate first-order Taylor expansion of $\boldsymbol{\xi}_K(\boldsymbol{\epsilon})$ around $\boldsymbol{\epsilon} = \mathbf{1}$ is:

$$\boldsymbol{\xi}_K(\boldsymbol{\epsilon}) \approx \boldsymbol{\xi}_K(\mathbf{1}) + \sum_{t_i \in S} \left. \frac{\partial \boldsymbol{\xi}_K(\boldsymbol{\epsilon})}{\partial \epsilon_{t_i}} \right|_{\boldsymbol{\epsilon}=\mathbf{1}} (\epsilon_{t_i} - 1) + o(||\boldsymbol{\epsilon}||)$$

Get $\boldsymbol{\xi}_K(\mathbf{0})$ with the approximation and plug it into the effect on the state:

$$\boldsymbol{w}_{K,S} = \boldsymbol{\xi}_K(\mathbf{1}) - \boldsymbol{\xi}_K(\mathbf{0}) \approx \sum_{t_i \in S} \left. \frac{\partial \boldsymbol{\xi}_K(\boldsymbol{\epsilon})}{\partial \epsilon_{t_i}} \right|_{\boldsymbol{\epsilon}=1} \tag{19}$$

The term $\left. \frac{\partial \boldsymbol{\xi}_K(\boldsymbol{\epsilon})}{\partial \epsilon_{t_i}} \right|_{\boldsymbol{\epsilon}=1}$ represents the first-order effect of step $t_i$. This effect is evaluated under the condition that all other steps in $S$ are skipped while all steps not in $S$ are executed. This term corresponds precisely to the definition of $\hat{\boldsymbol{w}}_{K,t_i}$ from Estimator 4.1. Here, the "base" trajectory for the single-step effect is one where $t_i$ is skipped but all other steps, including those in $S \setminus \{t_i\}$, are executed. The linearity of the Taylor expansion justifies this summation over all steps in $S$.

Therefore,

$$\hat{\boldsymbol{w}}_{K,S} = \sum_{t_i \in S} \hat{\boldsymbol{w}}_{K,t_i}$$

This proves the effect on the state in eq. (10). The proof for the behavior effect $\hat{\tau}_{K,S}$ follows directly as in App. A.1.

## B. Error Bound for Stage Effect Estimation

In this section, we derive an upper bound on the estimation error of the stage effect estimation.

First, we review the notations and restate Theorem 4.1 of the error bound for stage effect estimation from §4.3 for convenience.

- $\boldsymbol{\xi}_k \in \mathbb{R}^{2p}$: The state of the model at step $k$, comprising parameters $\theta_k$ and momentum/velocity $v_k$.

- $\eta_k$: The learning rate at step $k$.

- $K$: The last step in the model development process.

- $S$: A stage involves a set of update steps to be intervened on.

- $\boldsymbol{w}_{t_i+1,t_i}$: The initial effect on the state introduced by a single step $t_i$.

- $D_S := \sum_{t_i \in S} \|\boldsymbol{w}_{t_i+1,t_i}\|$: The cumulative magnitude of the initial effects on the state of the entire stage.

- $\mathbf{M}_k$: The propagator matrix at step $k$, capturing the model development dynamics.

- $\tau_{K,S}$: The true causal effect of stage $S$ at the final time $K$ on the performance $\gamma(\boldsymbol{x}, \boldsymbol{\theta}_K)$.

- $\hat{\tau}_{K,S}$: The first-order estimator of $\tau_{K,S}$.

**Theorem 4.1** (Error Bound for Stage Effect Estimation). *Assume the loss function $\mathcal{L}$ is twice differentiable with an L-Lipschitz continuous Hessian, and the optimization update maps are locally stable on the region between the observed and counterfactual trajectories, e.g., for the state update map $\Psi_k$, $\|D\Psi_k(\boldsymbol{\xi}_k)\|_2 \leq e^{\eta_k \Lambda}$. Let $\eta \geq \eta_k$ be the maximum learning rate. Let $D_S := \sum_{t_i \in S} \|\boldsymbol{w}_{t_i+1,t_i}\|$ be the magnitude of the initial effects on the state by stage $S$. Let the first and second derivatives of the performance function $\gamma$ be bounded. Then, the estimation error of the effect of stage $S$ on the final model performance is bounded by:*

$$|\tau_{K,S} - \hat{\tau}_{K,S}| \leq \frac{L_\gamma D_S^2}{2} e^{2\eta\Lambda(K-\min(S))} \tag{11}$$

The derived upper bound is governed by the interplay between the magnitude of the intervention, the geometry of the loss landscape, and the stability of the training dynamics:

$$|\text{Error}| \propto \underbrace{D_S^2}_{\text{Initial Effect on State}} \cdot \underbrace{L_\gamma}_{\text{Effective Constant}} \cdot \underbrace{e^{2\eta\Lambda(K-\min(S))}}_{\text{Training Regime}} \tag{20}$$

As discussed in §4.3, this bound characterizes the relationship between the approximation error and the properties of the loss landscape, the model development dynamics, and the magnitude of the initial effects on the state. It highlights three

regimes where the estimator may degrade: First, dependence on $D_S^2$ means that unusually large parameter updates can induce counterfactual trajectories that are too far from the observed trajectory for a first-order approximation. Second, a large Hessian Lipschitz constant $L$ indicates that local curvature changes rapidly, so the observed curvature becomes a weaker proxy for the counterfactual trajectory. Third, the exponential term becomes large under aggressive or unstable optimization, particularly for early interventions with many subsequent updates, during which errors can accumulate. Conversely, late-stage fine-tuning, moderate learning rates, and stable optimizer dynamics are the regimes where the estimator should be most reliable.

*Proof.* Write the full optimizer state as $\boldsymbol{\xi}_k = (\boldsymbol{\theta}_k, \boldsymbol{v}_k)$, and let

$$\delta_k := \boldsymbol{\xi}_k(1_S) - \boldsymbol{\xi}_k(0_S)$$

be the true state difference between the observed trajectory and the counterfactual trajectory that skips all steps in $S$. By definition, $\delta_k$ is exactly the true causal effect $\boldsymbol{w}_{k,S}$ of stage $S$ on the state at step $k$, whose first-order estimator $\hat{\boldsymbol{w}}_{k,S}$ we analyze below. For each skipped step, define the observed update increment

$$u_k := \boldsymbol{\xi}_{k+1} - \boldsymbol{\xi}_k.$$

Then

$$D_S = \sum_{t_i \in S} \|u_{t_i}\| = \sum_{t_i \in S} \|\boldsymbol{w}_{t_i+1, t_i}\|.$$

At a skipped step $k \in S$, the counterfactual state does not update while the observed state receives the update $u_k$, whereas at a non-skipped step $k \notin S$ both trajectories evolve under the same optimizer update map $\Psi_k$, so

$$\delta_{k+1} = \begin{cases} \delta_k + u_k, & k \in S, \\ \Psi_k(\boldsymbol{\xi}_k(1_S)) - \Psi_k(\boldsymbol{\xi}_k(0_S)), & k \notin S. \end{cases} \tag{21}$$

By the local stability assumption on the true update maps, the non-skipped case satisfies

$$\|\delta_{k+1}\| \leq e^{\eta_k \Lambda} \|\delta_k\|. \tag{22}$$

Let $t_1 = \min(S)$ be the earliest step in the stage. Unrolling eqs. (21) and (22), every update increment in $S$ is propagated through at most the remaining horizon after $t_1$. Thus, for all $k \leq K$,

$$\|\delta_k\| \leq D_S \exp\left( \Lambda \sum_{j=t_1+1}^{k-1} \eta_j \right) \leq D_S e^{\eta \Lambda (K - \min(S))}. \tag{23}$$

Define the stage linearized displacement $\hat{\delta}_k$. It starts at $\hat{\delta}_{t_1} = 0$ and evolves as

$$\hat{\delta}_{k+1} = \begin{cases} \hat{\delta}_k + u_k, & k \in S, \\ M_k \hat{\delta}_k, & k \notin S, \end{cases} \qquad M_k := D\Psi_k(\boldsymbol{\xi}_k(1_S)). \tag{24}$$

For non-skipped steps, Taylor expansion of $\Psi_k$ around the observed trajectory gives

$$\delta_{k+1} = M_k \delta_k + r_k, \qquad \|r_k\| \leq \frac{B}{2} \|\delta_k\|^2,$$

where $B$ is a second-order smoothness constant of the optimizer map. For SGD with momentum and weight decay on the state $(\boldsymbol{\theta}, \boldsymbol{v})$, the only nonlinear term is the batch gradient; if the loss Hessian is $L$-Lipschitz and $\eta_k \leq \eta$, we can take

$$B = \sqrt{1 + \eta^2} \, L. \tag{25}$$

Let $e_k := \delta_k - \hat{\delta}_k$. At skipped steps, both $\delta_k$ and $\hat{\delta}_k$ receive the same update increment $u_k$. At non-skipped steps,

$$e_{k+1} = M_k e_k + r_k.$$

Unrolling the error recursion, then substituting the sharper first inequality in eq. (23), bounding the exponent by $2\eta\Lambda(K - \min(S))$, and using that there are at most $K - \min(S)$ summands, gives

$$
\begin{aligned}
\|e_K\| &\leq \sum_{k=t_1+1}^{K-1} \exp\left(\Lambda \sum_{j=k+1}^{K-1} \eta_j\right) \frac{B}{2}\|\delta_k\|^2 \\
&\leq \sum_{k=t_1+1}^{K-1} \frac{B}{2}D_S^2 \exp\left(\Lambda \sum_{j=k+1}^{K-1} \eta_j + 2\Lambda \sum_{j=t_1+1}^{k-1} \eta_j\right) \\
&\leq \sum_{k=t_1+1}^{K-1} \frac{B}{2}D_S^2 e^{2\eta\Lambda(K-\min(S))} \\
&\leq \frac{B(K-\min(S))}{2}D_S^2 e^{2\eta\Lambda(K-\min(S))}.
\end{aligned}
\tag{26}
$$

Let $P_{\boldsymbol{\theta}}$ denote projection from the full state to the parameter block. Define the true final parameter displacement and its block-stage linearized estimate as

$$
\Delta\boldsymbol{\theta}_K := P_{\boldsymbol{\theta}}\delta_K, \qquad \hat{E}_S := P_{\boldsymbol{\theta}}\hat{\delta}_K.
$$

The first-order estimator of the performance effect is

$$
\hat{\tau}_{K,S} = \nabla_{\boldsymbol{\theta}}\gamma(\boldsymbol{x}, \boldsymbol{\theta}_K)^{\top}\hat{E}_S.
$$

Since the first and second derivatives of the performance function are bounded on the line segment between the observed and counterfactual final parameters, choose constants $G_1$ and $G_2$ such that

$$
\|\nabla_{\boldsymbol{\theta}}\gamma(\boldsymbol{x}, \boldsymbol{\theta})\| \leq G_1, \qquad \|\nabla_{\boldsymbol{\theta}}^2\gamma(\boldsymbol{x}, \boldsymbol{\theta})\|_2 \leq G_2
$$

on this segment. Define the effective constant $L_{\gamma} := G_1\sqrt{1+\eta^2}L(K - \min(S)) + G_2$.

Taylor expansion of the performance function around the observed final parameter gives

$$
\tau_{K,S} = \nabla_{\boldsymbol{\theta}}\gamma(\boldsymbol{x}, \boldsymbol{\theta}_K)^{\top}\Delta\boldsymbol{\theta}_K + \rho_{\gamma}, \qquad |\rho_{\gamma}| \leq \frac{G_2}{2}\|\Delta\boldsymbol{\theta}_K\|^2.
$$

Since projection cannot increase norm, $\|\Delta\boldsymbol{\theta}_K - \hat{E}_S\| \leq \|e_K\|$ and $\|\Delta\boldsymbol{\theta}_K\| \leq \|\delta_K\| \leq D_S e^{\eta\Lambda(K-\min(S))}$. Combining these with eq. (26),

$$
\begin{aligned}
|\tau_{K,S} - \hat{\tau}_{K,S}| &\leq G_1\|\Delta\boldsymbol{\theta}_K - \hat{E}_S\| + \frac{G_2}{2}\|\Delta\boldsymbol{\theta}_K\|^2 \\
&\leq G_1\|e_K\| + \frac{G_2}{2}\|\delta_K\|^2 \\
&\leq \frac{1}{2}\left(G_1 B(K - \min(S)) + G_2\right)D_S^2 e^{2\eta\Lambda(K-\min(S))} \\
&= \frac{1}{2}\left(G_1\sqrt{1+\eta^2}L(K - \min(S)) + G_2\right)D_S^2 e^{2\eta\Lambda(K-\min(S))} \\
&= \frac{L_{\gamma}D_S^2}{2}e^{2\eta\Lambda(K-\min(S))}.
\end{aligned}
\tag{27}
$$

$\square$

## C. Computational Considerations

### C.1. Computational Complexity and Structured Approximations

In this section, we derive the computational complexity of the full propagation for training stage effect estimation, as well as Hessian approximation and structured approximations for scaling.

**Complexity of full propagation** The propagation matrix $\mathbf{M}_k$ is of size $2p \times 2p$. Computing the overall propagator $\mathbf{P}^{((t+1)\to K)}$ involves approximately $K - t$ matrix-matrix multiplications. If each $\mathbf{M}_k$ is explicitly formed, each such multiplication costs $O((2p)^3) = O(p^3)$. Thus, forming $\mathbf{P}^{((t+1)\to K)}$ for a single step $t$ can be $O((K - t)p^3)$. An iterative algorithm can be used to compute all $E_t$ by updating a backward product, e.g., first computes $\mathbf{P}^{((K-1)\to K)}$ and then uses it to compute $E_{K-1}$. Then, update $\mathbf{P}^{((K-1)\to K)}$ to $\mathbf{P}^{((K-2)\to K)}$ by multiplying it with $\mathbf{M}_{K-2}$ and uses it to compute $E_{K-2}$, etc. The per-step cost in the backward pass involves a matrix-matrix product, leading to an overall complexity that can be roughly $O(Kp^3)$ or $O(K|\mathcal{B}|p^2)$ if Hessian-vector products are used efficiently within the matrix multiplication. The storage for $\mathbf{P}^{((t+1)\to K)}$ itself is $O(p^2)$.

**Structured approximations for scaling** The full propagation couples all $p$ parameters and is therefore difficult to scale directly. A common heuristic is to restrict the computation to the parameters of each layer $l$ (with dimension $p_l$) separately and then aggregate the resulting effects. This effectively assuming the independence between effects of different layers, and it is common in the literature with influence analysis of large models (Grosse et al., 2023). It reduces the computation to per-layer effect $E_t^l$ and changes the dominant complexity from $O(K|\mathcal{B}|p^2)$ to $O(K|\mathcal{B}|\sum_l p_l^2)$. If a model contains one or a few very large layers, those layers can be further partitioned into blocks, reducing the corresponding term from $p_l^2$ to $\sum_b p_{l,b}^2$, where $p_{l,b}$ is the parameter dimension of block $b$ within layer $l$. This block-wise variant trades off attribution fidelity for lower memory and runtime by dropping the interactions across blocks. A more aggressive option is to attribute only a subset of layers, such as the final prediction head or the LoRA adapter parameters used during fine-tuning. This reduces the complexity to $O(K|\mathcal{B}|p_{\text{sub}}^2)$ for the selected parameter subset. Our empirical observations, and those in related literature (Koh & Liang, 2017; Barshan et al., 2020), suggest that computations focused on prediction heads or adapter parameters can still yield stable and interpretable results when the target behavior is primarily mediated by those parameters. These structured approximations are not exact substitutes for the full estimator, but they provide a practical knob for scalability trade-off.

### C.2. Runtime and Memory Experiments

We further benchmark the practical runtime and memory overhead of AA-Score. As discussed in §4.4, the computational cost has two parts: (1) *logging*, which occurs during training and saves gradients and optimizer states, and (2) *attribution*, which is run after training to propagate perturbations. In Tables 3 and 4, time columns report wall-clock time in seconds. For AA-Score rows, each time entry is reported as logging/attribution time.

The logging overhead is larger for the ResNet results, approximately 3-4× for ResNet18 and 5-6× for ResNet34 relative to regular training, because I/O dominates these relatively small training runs. For Pythia LoRA fine-tuning, the required I/O is much lower (adapters only), so the logging overhead is approximately 1.1-1.2×. The attribution overhead is more manageable, ranging from a fraction of training time to roughly 0.7-1.8× training time in these settings. Also, it only needs to be computed once for all the stages are queried. Meanwhile, the runtime scales roughly linearly with the number of training examples and with the model size used in these benchmarks. For memory, peak memory is typically determined either by training or by the largest layer-wise attribution computation, and it is feasible unless there is one very large layer.

Table 5 shows that the estimated effects maintain reasonably high correlation with retraining counterfactuals across these model and dataset sizes. These results suggest that AA-Score is practical for small to medium-sized models on a single GPU, while scaling to production-scale foundation models remains an open challenge, but the structured approximations provide a path to trade attribution fidelity for scalability.

*Table 3.* Runtime and peak memory benchmarks on CELEBA with ResNet. Time is reported in seconds. For AA-Score, each entry is reported as logging/attribution time.

| Model | Mode | 2K | 4K | 8K | Peak Mem. |
|---|---|---|---|---|---|
| ResNet18 | Train | 288 | 460 | 824 | 350M |
| ResNet18 | AA-Score | 905 / 208 | 1582 / 406 | 2903 / 820 | 1.1G |
| ResNet34 | Train | 299 | 503 | 907 | 512M |
| ResNet34 | AA-Score | 1442 / 466 | 2908 / 850 | 5538 / 1610 | 2G |

*Table 4.* Runtime and peak memory benchmarks on CIVILCOMMENTS with Pythia LoRA fine-tuning. Time is reported in seconds. For AA-Score, each entry is reported as logging/attribution time.

| Model (LoRA params) | Mode | 2K | 4K | 8K | Peak Mem. |
|---|---|---|---|---|---|
| Pythia-70M (99K) | LoRA | 31 | 55 | 104 | 875M |
| Pythia-70M (99K) | AA-Score | 36 / 44 | 66 / 101 | 120 / 181 | 1.3G |
| Pythia-160M (296K) | LoRA | 88 | 164 | 312 | 2.4G |
| Pythia-160M (296K) | AA-Score | 101 / 129 | 183 / 248 | 346 / 448 | 2.4G |
| Pythia-410M (788K) | LoRA | 271 | 511 | 979 | 6.4G |
| Pythia-410M (788K) | AA-Score | 297 / 447 | 547 / 873 | 1049 / 1497 | 6.4G |

*Table 5.* Correlation between estimated stage effects and retraining counterfactuals in the runtime-scaling benchmarks. Each setting estimates one stage with 20 update steps.

| Benchmark | Model | 2K | 4K | 8K |
|---|---|---|---|---|
| CelebA | ResNet18 | 0.9231 | 0.9393 | 0.8207 |
| CelebA | ResNet34 | 0.9557 | 0.7607 | 0.9629 |
| CivilComments | Pythia-70M | 0.8665 | 0.9665 | 0.9899 |
| CivilComments | Pythia-160M | 0.9889 | 0.9941 | 0.9672 |
| CivilComments | Pythia-410M | 0.8814 | 0.9656 | 0.9632 |

# D. Datasets and Models

## D.1. Dataset Details

MNIST (LeCun, 1998): A standard benchmark dataset of handwritten digits. Due to its simplicity, it will allow for thorough case studies, including direct comparison with retraining to assess the accuracy of our approximation under various conditions (e.g., different inserted stages, different optimizers).

CELEBA (Liu et al., 2015): A large-scale face attributes dataset. This dataset is known to contain potential spurious correlations (e.g., gender with hair color). We aim to use our method to identify training stages where such spurious correlations might be predominantly learned by the model.

CIVILCOMMENTS (Borkan et al., 2019): A dataset of public comments labeled for toxicity and whether they contain words corresponding to demographic information like race, gender, and religion. This text dataset is often used for studying fairness and bias. We investigate if our method can identify training stages that disproportionately contribute to the model learning biases or relying on spurious correlations between certain identity terms and toxicity labels.

CHESTX-RAY14 (Wang et al., 2017), PADCHEST (Bustos et al., 2020), and VINDR-CXR (Nguyen et al., 2020): These are large-scale chest radiograph datasets collected from different geographic and demographic populations. CHESTX-RAY14 contains over 100,000 frontal-view X-rays from U.S. patients with text-mined labels for 14 thoracic pathologies. PADCHEST consists of more than 160,000 chest radiographs from Spain, annotated with radiographic findings, diagnoses, and anatomical locations, with a portion manually labeled by radiologists. VINDR-CXR is a Vietnamese dataset with expert bounding-box annotations for 14 thoracic abnormalities across nearly 20,000 chest X-rays. In our experiments, we specifically focus on the Cardiomegaly pathology, which is consistently annotated across all three datasets and clinically significant for assessing heart enlargement. Together, these datasets provide a diverse and clinically relevant benchmark for evaluating our method in medical imaging.

## D.2. Model Details

For each dataset, we employ model architectures appropriate to the task. For MNIST, we implement a three-layer MLP architecture with 128 hidden dimensions in each layer. This relatively simple architecture allows us to perform detailed analysis of training dynamics and enables direct comparison with retraining experiments, while still providing sufficient capacity to learn meaningful digit representations. For CELEBA, we utilize a ResNet-18 (He et al., 2016) model pre-trained on ImageNet as our backbone architecture. We augment this model with an additional final classification layer specifically trained to predict whether a celebrity has blonde hair. For the CIVILCOMMENTS dataset, we employ a pre-trained Gemma 3

model (Team et al., 2025) from the Huggingface library (Wolf et al., 2019) as our base architecture. We extend this model with a classification head trained to predict comment toxicity. For the chest X-ray datasets, we employ a DenseNet-121 (Huang et al., 2017) backbone architecture implemented within the TorchXRayVision (Cohen et al., 2020) framework. DenseNet-121 is a widely adopted model for medical imaging tasks due to its efficiency and strong performance in learning hierarchical visual features. We initialize the model with ImageNet-pretrained weights and finetune the final classification layer to predict the presence or absence of Cardiomegaly. This setup allows us to evaluate our method in a medical domain across datasets reflecting different patient demographics. The training hyperparameters are different for each experiment, which we specify in the following sections.

## E. Detailed Experiment Settings and Additional Results

### E.1. Retraining Counterfactual Protocol

For all datasets, retraining-based counterfactuals are used for evaluation. For most of our experiments with genuine multi-stage development, we first train the model regularly and save checkpoints at stage boundaries. These checkpoints are used as the observed state trajectory. To measure the ground-truth effect of a target stage, we remove that stage and resume the following stage from the last saved checkpoint immediately before the skipped stage. The following stage uses the same data, hyperparameters, and schedule as in the observed pipeline, with the momentum buffer and learning-rate scheduler reinitialized. This design mimics practical multi-stage development, where stages are separate training jobs initialized from previous checkpoints. For local anomaly experiments on MNIST, namely the inserted data point and mislabeled batches, the anomalous updates are not separate training runs. We therefore construct the counterfactual by running the same training run while skipping the specific parameter updates for those anomalous steps, mimicking the practical training process without the anomalous updates. The learning-rate scheduler still advances at skipped steps, but the momentum buffer is not updated because the skipped gradients are unavailable.

### E.2. Accountability Attribution on MNIST

#### E.2.1. Capture an Influential Positive Stage on MNIST

For this experiment, we train a 2-layer MLP with hidden dimension 128 on a small subset of MNIST of the first 100 samples, excluding digit '4'. We train for 1 epoch with batch size 1 using a learning rate of 0.001. We use a small subset and batch size 1 so we can insert a single instance of digit '4' at any training step. At training step 30, we insert a single instance of digit '4' from the test set to study its effect on itself, other images of digit '4', and other digits. In Fig. 3 (a), we show the effect of the training stages estimated by AA-Score. We can see that our attribution scores correctly identify the inserted step as having the highest positive effect on the model's performance on the same digit '4' classification. In Fig. 5, we further analyze the effect of inserting a test digit '4' during training on the model's ability to classify other digits, with Fig. 5 (a) being the same case as Fig. 3 (a) for effect on the same digit '4' that is inserted. For the other three plots, we pick another digit '4' from the test set different from the one inserted in Fig. 5 (b), a digit '9' which is easily confusable as the digit '4' in Fig. 5 (c), and a digit '2' which is visually distinct from '4' in Fig. 5 (d). We observe that the inserted step has the strongest positive effect on classifying other digit '4's, showing that the model learns generalizable features. The effect is slightly negative for digit '9', which shares some visual features with '4', suggesting learning the inserted digit '4' has negative effect on digit '9's classification. For digit '2', which is visually distinct from '4', the effect is close to neutral, slightly negative but not as large as the effect on '9', indicating that the learning is specific to relevant digit features.

#### E.2.2. Capture a Negative Stage Caused by Mislabeled Data on MNIST

For this experiment, we train a 2-layer MLP with hidden dimension 128 on a subset of MNIST of the first 10,000 samples for 1 epoch with batch size 100 and learning rate 0.001. We introduce label noise by flipping labels for 5% of the training samples. Specifically, starting from the 30th step, we modify labels of five consecutive batches (500 samples total) through a cyclic shift (digit $0 \rightarrow 1$, $1 \rightarrow 2$, etc.).

In Fig. 3 (b), we analyze the effect of these mislabeled training stages. Our method successfully identifies these stages as having significant negative effects on the model's test performance. The magnitude of negative effects correlates with the degree of label shift. This demonstrates our method's ability to quantify the harmful impact of noisy training data.

*Table 6.* The correlation of AA-Score for the specific stage and the counterfactual results obtained by retraining with the stage skipped. Results are computed over model performance on a randomly sampled test set. High correlation indicates that AA-Score successfully captures the effect of the stage. Stage 1, 2, and 3 are trained on CHESTX-RAY14, PADCHEST, and VINDR-CXR, respectively.

| Eval Dataset | Stage 1 | Stage 2 | Stage 3 |
|---|---|---|---|
| ChestX-ray14 | 0.9555 | 0.9890 | 0.7298 |
| PadChest | 0.9079 | 0.9872 | 0.6834 |
| VinDr-CXR | 0.9344 | 0.9665 | 0.8109 |
| Average | 0.9326 | 0.9809 | 0.7414 |

### E.2.3. ATTRIBUTE TO STAGES WITH DATA DISTRIBUTIONAL SHIFTS ON MNIST

For this experiment, we train a 2-layer MLP with hidden dimension 128 on a subset of MNIST of the first 2,000 samples with batch size 100 and learning rate 0.001. We introduce a distributional shift by rotating the images by 45 degrees in stage 2, and by 90 degrees in stage 3. We train for 3 epochs for stage 1, 1 epoch for stage 2, and 1 epoch for stage 3.

In Fig. 3 (c-e), we evaluate each stage's effect on three test sets: original orientation, 45-degree rotated, and 90-degree rotated. The results show clear specialization: each stage has maximum effect on its corresponding test distribution. For example, stage 2 (45-degree training) shows the highest positive effect on 45-degree rotated test images.

We also observe some transfer effects between stages. Training on 45-degree rotated images (stage 2) shows moderate positive effects on both 0-degree and 90-degree test sets, suggesting the model learns some rotation-invariant features. However, the 90-degree stage shows minimal positive effect on 0-degree test performance, indicating potential catastrophic forgetting of the original orientation when the distributional shift is too large.

### E.3. Accountability Attribution on CELEBA

For CELEBA experiments, we train the model on a subset of 4000 images from the dataset for 1 epochs with batch size 2 and learning rate 0.0005, momentum 0.9, and weight decay 0.00001. To investigate potential spurious correlations, we simultaneously evaluate the model's implicit learning of gender information on a balanced test subset across four demographic categories: *blonde-haired males*, *non-blonde-haired males*, *blonde-haired females*, and *non-blonde-haired females*, 100 images per category. This dual evaluation setup allows us to assess whether the model truly learns to classify hair color or if it relies on gender as a confounding variable in its decision-making process. For the 3-stage retraining evaluation experiment, we considered different training subset for each stage, 1000 images per stage, and evaluate on the same balanced test subset of 400 images.

### E.4. Accountability Attribution on CIVILCOMMENTS

For CIVILCOMMENTS experiments, we train the model on a subset of 5000 comments from the dataset for 1 epoch with batch size 50 and learning rate 0.0001, momentum 0.9, and weight decay 0.00001. To investigate potential biases, we specifically analyze the model's behavior regarding the confounding variable. For the spurious correlation removal experiment, we considered examples with the religious identity e.g., the terms "Christian" in the comments. This allows us to evaluate whether the model genuinely learns to classify toxicity or if it develops undesirable associations with specific demographic identifiers. For the 3-stage retraining evaluation experiment, we considered three different identity at 3 stages: religious identity "Christian" for stage 1, racial identity "black" for stage 2, and gender identity "male" for stage 3.

### E.5. Accountability Attribution on CHESTX-RAY14, PADCHEST, and VINDR-CXR

For the chest X-ray image experiments, we sampled 1000 images from each of the three datasets as training subsets. For stage 1, 2, and 3 we use subset from CHESTX-RAY14, PADCHEST, and VINDR-CXR, respectively. The training for each stage is for 2 epochs with batch size 50 and learning rate 0.01, momentum 0.9, and weight decay 0.00001. For the 3-stage retraining evaluation, we randomly sample a test subset of 200 images from each of the three chest X-ray datasets (CHESTX-RAY14, PADCHEST, and VINDR-CXR) and compute the correlation between our method's predicted stage effect and the counterfactual results obtained by retraining with the corresponding stage skipped. For the retraining results, we report the correlation of test data separately sampled from each X-ray dataset in table Table 6. The correlation reported in the main paper is the average over each column.

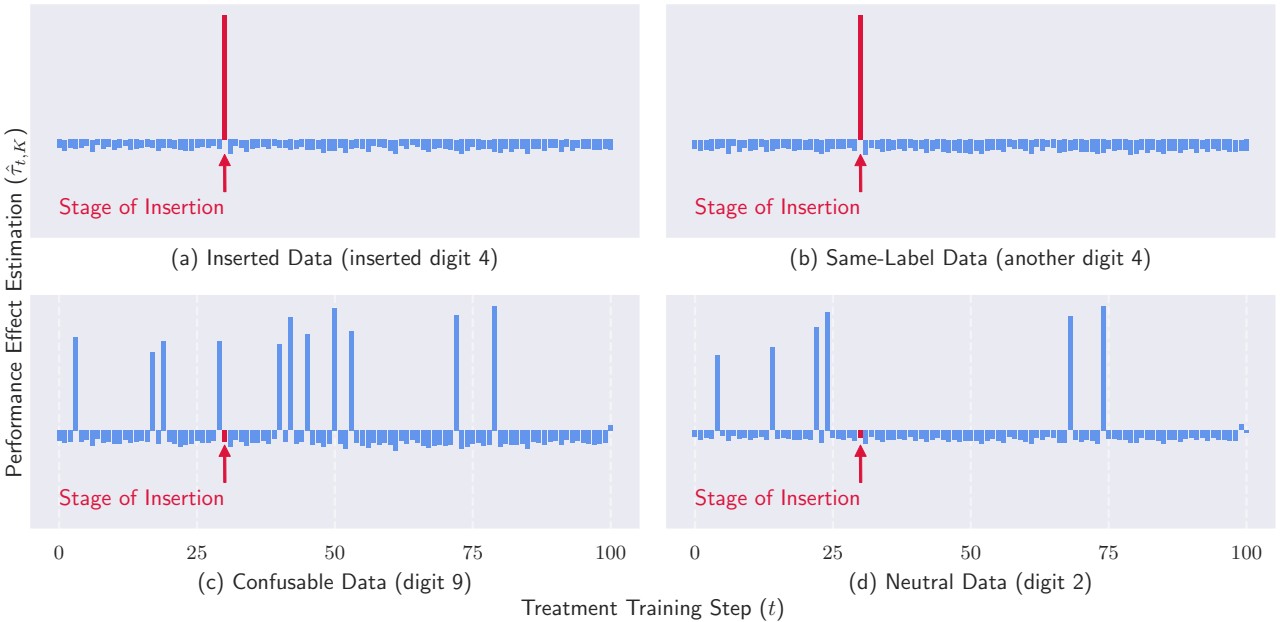

*Figure 5.* The effect of inserting a test digit '4' during training on the model's ability to classify four different digits. (a) is the same case as Fig 1 (a) for effect on the same digit '4' that is inserted. (b) is the effect on another digit '4' from the test set. (c) is the effect on digit '9', which is easily confusable as '4'. (d) is the effect on a neutral digit '2', which is visually distinct from '4'.

### E.6. Accountability Attribution on Dispersed Backdoor Attacks

We evaluate AA-Score on BadNets (Gu et al., 2019) backdoor attacks following a standard setting in the BackdoorBench (Wu et al., 2022) on CIFAR-10 with ResNet18. This setting tests whether AA-Score can identify malicious training signals when poisoned examples are not isolated as one obvious contiguous stage. In Fig. 6, poisoned examples are inserted into dispersed batches. Each batch contains either regular or poisoned examples, and AA-Score clearly assigns poisoned batches positive contributions to test examples with the target label and negative contributions to test examples with non-target labels. In Fig. 7, poisoned examples are fully dispersed so every batch contains a mixture of regular and poisoned examples. This more challenging setting no longer yields a simple binary distinction between poisoned and clean batches. Although the target-label and non-target-label attribution patterns remain different, showing that AA-Score can expose backdoor-related training effects even when the poisoning signal is spread across the training process, AA-Score still loses its discriminative power as in the isolated stage setting. For such extreme cases, data-point-level attribution methods are more appropriate.

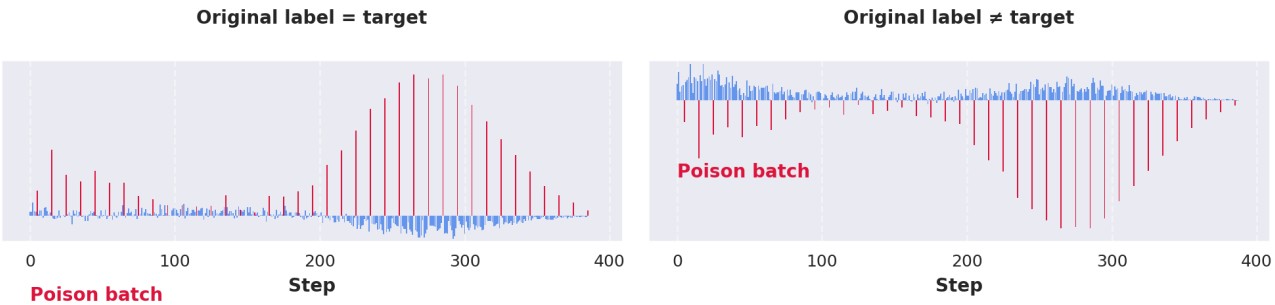

*Figure 6.* AA-Score results for a BadNets backdoor attack on CIFAR-10 with ResNet18 following BackdoorBench. Poisoned data are inserted during training in dispersed batches. Each batch contains either regular examples (blue) or poisoned examples (red). Batches with poisoned examples are clearly identified as making positive contributions to test examples with the target label (left) and negative contributions to test examples with non-target labels (right).

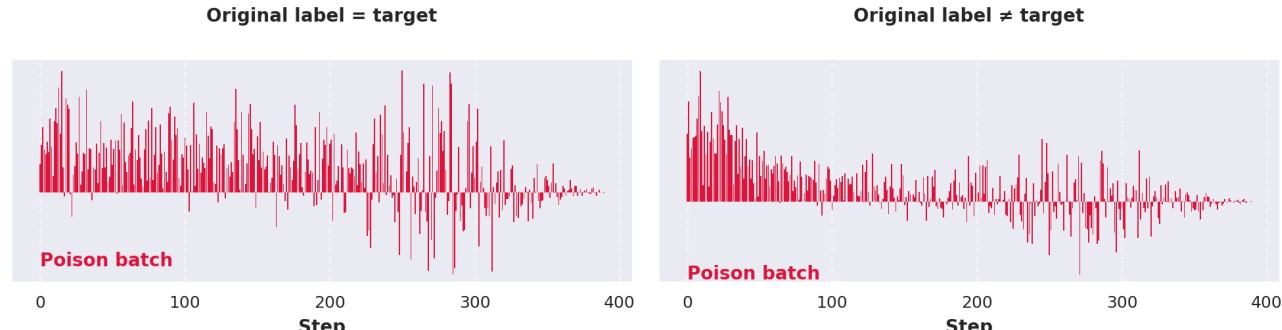

*Figure 7.* AA-Score results for a BadNets backdoor attack on CIFAR-10 with ResNet18 following BackdoorBench. Poisoned data are fully dispersed during training, so each batch contains both poisoned and regular examples (all batches shown in red). Batch contributions can be either positive or negative. Contributions to test examples with the target label (left) and non-target labels (right) show different patterns.

