# OpenReview forum: "Who Gets Credit or Blame? Attributing Accountability in Modern AI Systems"
_ICML.cc/2026/Conference — ICML 2026 regular_

### Official Review · Reviewer_TGAG · 2026-02-17

**Soundness:** 3
**Presentation:** 3
**Significance:** 2
**Originality:** 2
**Overall Recommendation:** 4
**Confidence:** 3

**Summary:**

This paper studies stage-level accountability in multi-stage training and asks which part of the training process should receive credit or blame for a model outcome. It formalizes the question as a counterfactual effect of removing a chosen training stage and proposes a first-order estimator that propagates the influence of skipped updates through the remaining optimization dynamics. The method produces an AA-Score for each stage and can be used to audit which stages increase reliance on spurious signals. Experiments cover controlled MNIST settings and several datasets including CelebA, CivilComments, and chest X-ray datasets.

**Compliance With Llm Reviewing Policy:**

Affirmed.

**Final Justification:**

This paper presents a novel perspective on attributing model behavior to stages of the training process. I find the problem formulation and the core idea promising. The empirical results, while still mostly in controlled settings, provide reasonable support for the proposed framework. Although there are limitations in computational cost, scalability, and broader real-world applicability, I believe the paper makes a worthwhile contribution overall.

**Key Questions For Authors:**

1. What prior knowledge is needed to define stages, and in what real scenarios would this be available?

2. What is the wall-clock overhead in hours or percent slowdown during training?

3. How do runtime and memory scale with model depth, parameter count, and dataset size?

**Limitations:**

yes

**Strengths And Weaknesses:**

**Strengths**

1. The goal is novel and practically motivated. It targets auditing and attribution of training stages and can help diagnose data-induced biases in a trained model.

2. The presentation is clear. The paper defines the counterfactual target precisely and the estimator is explained in a structured way.

3. The method is technically interesting and different from standard data attribution. It provides a post-hoc way to estimate how removing a stage would change downstream performance without retraining for every stage.

**Weaknesses**
1. The scope appears overclaimed, and the practical applicability is limited without strong prior knowledge. The approach requires a user-defined notion of stage, and in many realistic pipelines the analyst must already have a hypothesis about which data or which training segment is suspicious. With large datasets and stochastic sampling, it is difficult to map the suspected factor to a reliable set of steps, and the method does not actively discover problematic data when such priors are missing.

2. The paper proposes computing AA Scores along the training trajectory and discusses computational complexity, but it does not provide empirical evidence on the impact on training efficiency. There is no wall clock overhead or GPU hour reporting for representative settings, which makes it hard to judge feasibility in the model training process.

3. The experimental evidence is not fully convincing for the intended high-stakes auditing narrative. Much of the mechanistic validation relies on MNIST and on carefully constructed or benchmark spurious correlation settings. Stronger stress tests would be more appropriate, such as backdoor scenarios and adversarial training, to evaluate whether the estimator remains reliable under more severe failure modes.

4. Computational scalability remains under-discussed in practice. The cost depends on the training data and the model dimension through propagation. The paper would benefit from a systematic study that varies model size and dataset size and reports how runtime and memory scale, including the tradeoffs of the proposed approximations.

---

> ### Author Rebuttal · Authors · 2026-03-31
>
> We deeply appreciate your constructive feedback and address your questions below.
>
> >### Q1. Prior knowledge to define stages
>
> We view the flexibility to define stages as an advantage, not a weakness. Our framework supports multi-resolution analysis depending on the available information and the auditor's needs.
>
> - If no prior knowledge, we can treat each training batch as a "stage." This allows for fine-grained auditing as in the MNIST sanity checks with inserted data, similar to batch-level data attribution but accounting for complex optimization dynamics.
> - In real-world scenarios, stages are often predefined by the development pipeline (e.g., pre-train + fine-tune). Stages may also correspond to naturally separate data sources. For example, in the X-ray experiment, stages represent hospitals with patients from different demographic groups where data sharing is challenging. This natural prior is used without violating privacy. Our tool enables auditors to assign accountability to these natural stages.
>
> By allowing users to set the granularity, our method can be used for both finer and high-level auditing.
>
>
> >### Q2 training overhead and Q3 runtime and memory study
>
> We first clarify the computational cost mentioned in Q2 and Q3, and then report new experiment results, and then discuss these results to address the questions.
>
> The cost consists of two parts: Logging (during training) and Attribution (after training). Logging (during training) and Attribution (after training). Logging saves gradients and optimizer states; the primary time stems from I/O operations. Attribution corresponds to our proposed AA algorithm for perturbation propagation (theoretical complexity analyzed in App C).
>
> Given limited rebuttal time, we pick two tasks for new experiments: A CV task on CelebA with ResNet18 & ResNet34 for 2K, 4K, and 8K data. An NLP task on CivilComments with Pythia70M, Pythia160M, & Pythia410M for 2K, 4K, and 8K data. Below, each row reports the wall-clock time (s) and peak memory for regular training or AA. For AA rows, **Logging/Attribution** time is both reported.
>
> |Model|Mode|Time(2K)|Time(4K)|Time(8K)|Peak Mem|
> |-|-|-|-|-|-|
> |ResNet18|Train|288|460|824|350M|
> ||AA|905/208|1582/406|2903/820|1.1G|
> |ResNet34|Train|299|503|907|512M|
> ||AA|1442/466|2908/850|5538/1610|2G|
>
> |Model(#LoRA Param)|Mode|Time(2K)|Time(4K)|Time(8K)|Peak Mem|
> |-|-|-|-|-|-|
> |Pythia-70M(99K)|LoRA|31|55|104|875M|
> ||AA|36/44|66/101|120/181|1.3G|
> |Pythia-160M(296K)|LoRA|88|164|312|2.4G|
> ||AA|101/129|183/248|346/448|2.4G|
> |Pythia-410M(788K)|LoRA|271|511|979|6.4G|
> ||AA|297/447|547/873|1049/1497|6.4G|
>
> >#### Q2
>
> The training overhead comes from logging. For ResNet, it is ~3-4x (RN18) and ~5x-6x (RN34) to regular training. For Pythia LoRA, it is ~1.1-1.2x. The overhead mainly comes from I/O and is handled asynchronously. When training computation (e.g., backprop) dominates I/O, it becomes negligible (e.g., LoRA cases). We think there is plenty of room for further optimization using multi-processing I/O, which we consider as a future direction. On high-performance machines that clear the I/O bottleneck, this procedure for ResNet can be much faster without better GPUs.
>
>
> >#### Q3
>
> Compared to the logging time, our attribution time is more manageable. It ranges from ~0.7x to ~1.8x of the training time, which is preferable compared to retraining attribution, especially because it only needs to be done once, even when we want to consider many stages. Meanwhile, the attribution time scales roughly linearly with model size (e.g., RN34 doubles RN18 time for the same data size), and data size (e.g., 2K->4K->8K; each attribution time doubles for the same model).
>
> The peak memory of AA may be achieved either during logging or attribution, as we use a layer-wise approximation for attribution (App C). If there is one large layer, its attribution may become the peak (ResNet & Pythia-70M). Otherwise, the memory cost of regular training is the peak (Pythia-160M & 410M).
>
> We also show the corresponding attribution correlation for these settings for estimating one stage with 20 update steps. We see reasonably high correlation is maintained across model and data sizes.
>
> ||2K|4K|8K|
> |-|-|-|-|
> |ResNet18|0.9231|0.9393|0.8207|
> |ResNet34|0.9557|0.7607|0.9629|
>
> ||2K|4K|8K|
> |-|-|-|-|
> |Pythia-70M|0.8665|0.9665|0.9899|
> |Pythia-160M|0.9889|0.9941|0.9672|
> |Pythia-410M|0.8814|0.9656|0.9632|
>
> Overall, we think the runtime and memory are manageable for medium-sized experiments, e.g., from a few to 20-ish minutes. Also, given the scaling trend, we think the method has potential to scale up. While large-scale experiments exceed our current compute and rebuttal time constraints, we acknowledge this in the limitations section and identify structured approximations as future work to bridge this gap.
>
> **We hope these responses address your concerns, and we hope you consider raising your score.**

---

> > ### Author Rebuttal · Reviewer_TGAG · 2026-04-02
> >
> > I appreciate the authors’ effort in providing additional results. These results partially address my concern regarding the training and explanation overhead. I also agree that the goal of attributing accountability across training stages is promising. However, I still have some questions and concerns about the current application scope of the method.
> >
> > 1. My understanding is that the method still assumes the model's misbehavior is already known, so that one can define an appropriate auditing metric. In practice, this requires prior knowledge of the issue, a well-designed indicator, and a model training. Under this setting, I am still unclear about the practical advantage of this framework beyond only data or training-stage inspection. More specifically, compared with existing approaches for model correction, robust training, or debiasing, what are the main practical advantages or benefits of this method?
> > 2. At present, the explored scenarios still appear manually constructed or limited to relatively controlled settings, such as MNIST and the staged X-ray experiments. This raises two related questions for me. **First**, do the authors believe the method can remain effective for more subtle or realistic issues, such as backdoor attacks, realistic label noise, or naturally occurring data imbalance? If so, would success in these settings require much finer stage granularity and therefore substantially higher computational cost? **Second**, many practical training pipelines involve random sampling, and undesirable behavior such as spurious correlation or distribution shift may arise from examples dispersed throughout training rather than from a cleanly isolated stage. In such cases, what insights do the authors have about the applicability of their framework?
> >
> > Due to the short discussion time, I do not expect additional experiments, but I would appreciate a more explicit discussion of these limitations and the intended scope of the method.

---

> > > ### Author Response · Authors · 2026-04-06
> > >
> > > We appreciate your constructive feedback.
> > >
> > > >### Q1. Prior knowledge to define stages
> > >
> > > The core distinction between our method and the mitigation methods suggested by the reviewer is that we focus on **quantitative diagnosis**, whereas model correction, robust training, and debiasing focus on the **cure**. We believe that quantifying the accountability of a misbehavior is orthogonal and equally important to mitigating it, offering unique practical value.
> > >
> > > First, our method applies to unknown misbehaviors, meaning prior knowledge of a specific misbehavior is not strictly required. Using a general metric like validation loss, our method acts as an exploratory diagnostic tool to identify which stages drive overall success or failure. When no clear misbehavior exists, accountability can translate to assigning credit for positive performance contributions.
> > >
> > > Second, for known misbehaviors (e.g., spurious correlations), the reviewer is correct that a specific auditing metric is utilized. However, mitigating a misbehavior and quantifying its root cause remain distinct challenges. As discussed in related work, assessing accountability requires careful causal consideration (Halpern & Kleiman-Weiner, 2018). Our method bridges the gap in applying these causal concepts to the multi-stage training of AI models.
> > >
> > > A practical benefit of accountability attribution lies in multi-stakeholder AI development. Quantifying each stage's contribution provides concrete evidence for **liability tracking or profit sharing**. For example, a recent work by Laufer et al., "Bargaining and Adaptation for General-Purpose Models," studies model adaptation as a multi-stage bargaining game between developers. Their analysis assumes each stage's contribution is known. In reality, isolating the exact value added by each stage can be complex. Our method provides the technical pre-step to quantify these contributions, allowing such economic and legal frameworks to be applied.
> > >
> > > Furthermore, in a decentralized AI pipeline, a regulator or downstream company often lacks the control to force upstream companies to apply mitigation methods. In these cases, a centralized "cure" can be impossible; quantifying accountability is the necessary first step, which clarifies the responsible party and enables more precise and efficient mitigation.
> > >
> > > We will expand our discussion section to highlight these connections and differences.
> > >
> > >
> > > >### Q2 Subtle or realistic issues
> > >
> > > We do believe the proposed method is effective for some more subtle and realistic issues, but it certainly also has limitations.
> > >
> > > - Backdoor attacks and examples dispersed throughout the training
> > >
> > > The mechanism of a backdoor attack with poisoned data is analogous to our data insertion and mislabeling experiments (Fig 3a & 3b in the paper), where the anomalous stage was clearly identified. To provide concrete evidence, we ran a BadNets backdoor attack on CIFAR-10 using ResNet-18 (with ImageNet pre-trained weights) using the BackdoorBench library (Wu et al., 2022), and then we applied our attribution to quantify the accountability. We do attack in two settings with resulting **New Figures** in https://tinyurl.com/yb6rhpt6
> > >
> > > As in **New Figure 1**, our method can successfully identify poisoned batches dispersed throughout the training process. It quantified the positive/negative contribution these batches made to predicting the target/non-target labels.
> > >
> > > We also acknowledge a limitation regarding extreme dispersion. Since our method is built for stage-level attribution, its minimum unit of analysis is a single model update (one batch). If an issue is fully dispersed, so that every batch contains a mix of regular and poisoned data, our method loses its discriminative power (as in **New Figure 2**). For such extreme cases, data-point-level attribution methods are more appropriate.
> > >
> > > - Realistic label noise and data imbalance
> > >
> > > The effectiveness of our method in these scenarios depends on the specific data distribution. For systematic label errors at the stage/batch-level, our method can track the responsible stage, similar to our mislabeled experiments (Fig 3b in the paper). For natural data imbalance, we note that the CelebA dataset evaluated in our paper is imbalanced, and the successful identification of spurious correlations demonstrates the method's practical potential for such cases.
> > >
> > > - Computational cost.
> > >
> > > Analyzing these granular issues does not increase computational cost. As shown in App C, the complexity is $O(K|\mathcal{B}|p^2)$, which depends strictly on the total number of training steps $K$. Redefining stage granularity or evaluating individual batches adds no additional computational overhead.
> > >
> > > We will add a dedicated discussion in the revision to explicitly clarify the boundary between batch-level and data-point-level attribution.
> > >
> > > **We really appreciate your constructive feedback, and we hope our responses and planned revisions encourage you to consider raising your score.**

---

### Official Review · Reviewer_hxWq · 2026-03-11

**Soundness:** 3
**Presentation:** 4
**Significance:** 3
**Originality:** 3
**Overall Recommendation:** 5
**Confidence:** 3

**Summary:**

This paper studies how to attribute responsibility for model behavior across different stages of the training pipeline. The authors formulate this as a counterfactual problem, asking how the model’s behavior would change if a particular training stage had not occurred. They propose an efficient estimator, called AA-Score, that approximates these stage-level effects using first-order approximations of the training dynamics, avoiding the need to retrain the model.

**Compliance With Llm Reviewing Policy:**

Affirmed.

**Final Justification:**

I remain positive about the paper: it proposes a clear and well-motivated framework for attributing responsibility across training stages, with solid theoretical grounding and convincing empirical results. The rebuttal clarified scalability and approximation reliability, adequately addressing my main concerns, so I maintain my recommendation.

**Key Questions For Authors:**

1. The experiments are conducted on relatively moderate-scale models and training pipelines. How do the authors envision the approach scaling to modern large-scale training setups, such as foundation models or LLM pipelines that involve many stages (e.g., pretraining, instruction tuning, alignment, etc.) and long training trajectories?
2. The method relies on a first-order Taylor approximation of the training dynamics. While the small-scale retraining comparisons in the paper show encouraging correlations, it is not entirely clear when this approximation should be expected to remain reliable. Could the authors provide more intuition or empirical evidence about the regimes where the approximation breaks down (e.g., large updates, highly nonlinear training phases)?

**Limitations:**

Yes.

**Strengths And Weaknesses:**

**Strengths:**
1. The paper is very well written and presents the problem and the motivation in a very clear and engaging way.
2. The paper identifies an important but relatively underexplored question: how to attribute responsibility for model behavior across different stages of the training pipeline. As multi-stage training has become standard practice in modern AI systems, understanding which stage introduced a particular behavior is practically relevant for debugging, auditing, and improving models.
3. The idea of framing the problem through the lens of causal inference and counterfactual reasoning is compelling. Modeling training stages as interventions and defining stage effects via counterfactual outcomes provides a principled and intuitive formulation of accountability attribution.

**Weaknesses:**
1. The empirical evaluation is limited to moderate-scale settings, and I would have liked to see the method tested on more diverse and challenging training pipelines, especially closer to modern large-scale foundation models or LLM training. As it stands, it is still not very clear how well the approach would hold up in those more realistic settigns.
2. the first-order approximation seems reasonable, and the small-scale retraining comparisons do give some support for it, but I am still not fully convinced it is enough in strongly nonlinear or larger-scale settings. The paper does not yet establish when this approximation should be expected to remain reliable in realistic modern training pipelines.

---

> ### Author Rebuttal · Authors · 2026-03-31
>
> We deeply thank the reviewer for the strong support of our paper and for the constructive feedback. We address your specific questions below.
>
> >### Q1 Scaling up to large-scale training setups such as foundation models
>
> We thank the reviewer for highlighting this critical direction. We entirely agree that applying accountability attribution to large-scale LLM pipelines (e.g., pre-training $\rightarrow$ instruction tuning $\rightarrow$ alignment) is a highly practical and important application of our framework. As discussed in Sec. 6, this remains a limitation of our current empirical scope, with Gemma-3-1B being the largest model evaluated. To the best of our knowledge, no similar attribution methods currently scale to production-level LLMs. While conducting full-scale LLM training experiments is beyond our current compute and rebuttal time constraints, we envision a viable pathway to scale up through modular attribution for dimensionality reduction, which we are actively exploring.
>
> Specifically, as detailed in App. C, the primary computational bottleneck is tracking the full parameter dimension $p$, leading to a propagation complexity of $O(K|\mathcal{B}|p^2)$. By assuming layer-wise independence, this complexity reduces drastically from $O(p^2)$ to $O(\sum p_l^2)$, making the tracking of optimization dynamics far more computationally feasible. For models with excessively large layers, we can aggressively partition layers into smaller blocks, further reducing complexity to $O(\sum_l \sum_{block} p_{block}^2)$. Alternatively, computation can be restricted to specific layers (e.g., the final prediction head), a standard practice in the data attribution literature. These approximations allow us to strategically trade off attribution accuracy for scalability.
>
> Another potential bottleneck for scaling up is the reliability of the training dynamics approximation, as the reviewer pointed out in Q2, which we discuss below.
>
>
> >### Q2 Reliability of the training dynamics approximation
>
> We appreciate the reviewer's request for deeper intuition regarding the limitations of the training dynamics approximation. Our theoretical error bound (Theorem 4.1) characterizes the operational boundaries of our estimator. The estimation error is bounded by $\frac{LD_S^2}{2\Lambda}e^{2\eta\Lambda(K-t_1)}$. Based on this bound, our approximation accuracy is expected to degrade in the following specific regimes:
>
> Large updates: The error scales with the square of the counterfactual intervention magnitude, $D_S^2$. If the skipped step corresponds to a massive update, the counterfactual trajectory diverges too far from the observed one, decreasing estimation accuracy. Empirically, we observe that if an unusually large learning rate is used (e.g., 0.1), the estimation is off and the correlation with the retrained ground truth degrades. Fortunately, for reasonable learning rates used in practice, the estimation maintains a strong correlation.
>
> Highly nonlinear loss landscape: The error scales linearly with $L$, the Lipschitz constant of the Hessian. In highly non-convex regions of the loss landscape, the observed curvature becomes a poor predictor of the counterfactual path, leading to reduced estimation accuracy.
>
> Optimizer stability: The $2\eta\Lambda(K-t_1)$ term indicates that the error depends on the optimizer instability constant $\Lambda$ and learning rate $\eta$. The approximation's accuracy is sensitive to how aggressive the optimizer is. A larger spectral bound $\Lambda$ amplifies the initial discrepancy between the observed and counterfactual trajectories more rapidly. This instability is most pronounced during early-stage interventions, where the longer remaining training horizon $K-t_1$ allows these errors to accumulate.
>
> We will expand Section 6 and Appendix B to include this explicit intuition to better clarify when practitioners should trust the approximation. Despite these theoretical boundaries, we emphasize that in standard model development pipelines, these extreme regimes are rarely encountered. Once encountered, the model training will most likely break. Common optimization practices, such as learning rate schedules, warmup phases, and proper initialization, etc, naturally constrain update magnitudes and stabilize the trajectory. Consequently, for the majority of practical multi-stage pipelines, especially mid-to-late stage fine-tuning and domain adaptation, these theoretical concerns are minimal. The first-order approximation remains a reliable tool for accountability attribution.
>
> We sincerely thank the reviewer for the support of our work. We are actively working on scaling up this method to large foundation models.

---

> > ### Author Rebuttal · Reviewer_hxWq · 2026-04-03
> >
> > Thank you for the rebuttal and clarifications. I will be maintaining my score of 5.

---

> > > ### Author Response · Authors · 2026-04-06
> > >
> > > We appreciate the reviewer’s support of our paper and the constructive feedback that has helped improve its quality.

---

### Official Review · Reviewer_VT5w · 2026-03-11

**Soundness:** 3
**Presentation:** 3
**Significance:** 3
**Originality:** 3
**Overall Recommendation:** 5
**Confidence:** 3

**Summary:**

The paper introduces the "accountability attribution" problem, which traces an AI model’s behaviors back to specific stages of its development pipeline (e.g., pre-training, fine-tuning, alignment) . The authors formalize this using the potential outcomes framework, defining a stage’s causal effect as the counterfactual difference in model behavior if that stage’s updates had been skipped.
To validate the method, the authors establish ground truth by retraining counterfactual models where a specific stage is skipped . Specifically, they execute the training pipeline as normal but bypass all update steps for the target stage (applying subsequent stage updates directly to the previous stage's weights) and measure the resulting performance difference.
To estimate this effect without the high cost of retraining, the authors derive the Accountability Attribution Score (AA-Score). This uses a first-order Taylor approximation to model the optimization trajectory, explicitly accounting for learning rate, momentum, and weight decay.
Experiments across MNIST, CelebA, CivilComments, and medical X-ray datasets show that AA-Scores achieve good correlation (often 0.94+) with the retraining ground truth. The framework effectively identifies which stages are responsible for both beneficial learning and harmful behaviors, such as spurious correlations, domain generalization, and label noise.

**Compliance With Llm Reviewing Policy:**

Affirmed.

**Final Justification:**

The paper shows promise and focuses on an important topic. While I am not an expert in this specific sub-area, based on considering also the other reviewers' feedback and the rebuttal, I am prepared to raise the score from Weak Accept to Accept.

**Key Questions For Authors:**

1. How does the AA-Score quantitatively compare to existing process-based attribution methods (e.g., TracIn, DVEmb) on stage-level tasks? Including these baselines would help contextualize the proposed method's relative performance and strengthen the overall empirical evaluation.
2. When skipping a stage to compute the ground truth, subsequent optimization states (e.g., momentum, LR schedules) are not adjusted. Could you clarify whether this approach represents a realistic alternative development path, or if it risks inducing artificial optimization failures? Further discussion on this point would help establish the robustness of the counterfactual trajectories used to compute the ground-truth metrics.
3. Could you provide concrete runtime and memory consumption benchmarks for computing the AA-Score in your current experiments? Additionally, how do you project these costs scaling for larger foundation models? Providing concrete resource profiles would help practitioners better understand the method's scalability and its practical applicability to larger architectures.

**Limitations:**

yes.

The authors adequately discuss limitations including first-order approximation constraints, computational complexity challenges for large-scale models, and applicability questions to foundation model settings. The limitation section appropriately identifies future research directions around higher-order approximations, structured approximations, and distributed computing approaches. However, the negative societal impact discussion could be expanded with more explicit consideration of how attribution results might be misused or misinterpreted in accountability contexts (e.g., potential for cherry-picking stage-level evidence to shift blame).

**Strengths And Weaknesses:**

**Soundness**

- **Strengths:** The paper introduces a sound theoretical framework for attributing model behaviors to specific stages of a multi-stage AI development pipeline by utilizing counterfactual reasoning. Grounded in causal inference techniques, the method provides a logically consistent approach to estimating stage effects. The empirical validation show practical validity across diverse domains (vision, language, medical X-rays), achieving good correlations (e.g., 0.94+ on MNIST) with ground-truth retraining counterfactuals.
- **Weaknesses:** The experimental evaluation currently lacks direct quantitative comparisons with existing process-based data attribution methods (such as the cited TracIn or DVEmb), making it difficult to fully contextualize the performance gains. Furthermore, the definition of the ground truth counterfactual (which skips a target stage's updates without adjusting subsequent learning rates or optimization states) requires further justification to confirm it represents a realistic alternative development path rather than a mathematical construct that might induce feature collapse. Finally, the submission would benefit from additional ablations examining attribution stability across different random seeds and initialization variations.

**Presentation**

- **Strengths:** The paper is generally well-structured and clearly introduces the "accountability attribution" problem. The narrative effectively bridges the gap between data attribution and learning dynamics, and the integration of optimization factors is explained clearly.
- **Weaknesses:** While the paper claims efficiency advantages over retraining, it lacks concrete runtime and memory benchmarks for the models evaluated (MLP, ResNet-18, fine-tuned Gemma-3). Including these metrics is necessary to clarify the exact computational overhead and allow practitioners to properly assess the method's efficiency.

**Significance**

- **Strengths:** The paper addresses a relevant and timely problem in modern machine learning: accountability in multi-stage AI pipelines. By pinpointing the specific training stages responsible for both harmful behaviors (spurious correlations, label noise) and beneficial learning signals (domain generalization), the method provides a valuable tool for auditing and debugging without the prohibitive cost of full retraining.
- **Weaknesses:** While acknowledging computational limitations, the method's inherent complexity poses scaling challenges for large foundation models. If the method cannot be practically scaled or approximated for these larger models, its overall significance to the current foundation model paradigm may be constrained.

**Originality**

- **Strengths:** To the best of my knowledge, the work introduces a novel problem formulation by shifting the focus of attribution from individual data points to entire training phases. Furthermore, explicitly modeling optimization factors like learning rate schedules, momentum, and weight decay within the attribution estimator represents a practical advancement.

---

> ### Author Rebuttal · Authors · 2026-03-31
>
> >### Q1 Baseline (DVEmb)
>
> We implemented a stage-level version of DVEmb. Since DVEmb was designed for individual data points, we adapted it by first aggregating over data in a batch, and then across batches in a stage. We compare DVEmb to AA-Score under the settings of Table 1&2 in the paper.
>
> Table 1
> ||Method|Insert|Mislabel|S1|S2|S3|Avg|
> |-|-|-|-|-|-|-|-|
> |MNIST|DVEmb|0.9441|**0.9496**|**0.9521**|0.9329|0.9297|0.9417|
> ||AA|**0.9444**|0.9487|0.9518|**0.9350**|**0.9482**|**0.9456**|
>
> Table 2
> ||Method|S1|S2|S3|
> |-|-|-|-|-|
> |CelebA|DVEmb|0.4406|0.6304|0.4203|
> ||AA|**0.5524**|**0.6953**|**0.8436**|
> |Civil|DVEmb|0.7766|0.9569|0.8514|
> ||AA|**0.8576**|**0.9789**|**0.9746**|
> |X-Ray|DVEmb|**0.9628**|0.9487|0.7025|
> ||AA|0.9326|**0.9809**|**0.7414**|
>
> We see DVEmb is comparable to AA on MNIST, but it is outperformed on complex settings, except X-Ray S1. We identify 2 primary reasons why AA outperforms in the multi-stage context
>
> 1. The complex training dynamics are not entirely captured by DVEmb as it assumes SGD dynamics. Multi-stage training can further involve sharp transitions like velocity re-initialization. AA, by contrast, closely models such dynamics.
> 2. DVEmb is naturally data-centric, whereas AA is built for stage-level accountability. Although we tried our best to reproduce DVEmb faithfully for the multi-stage setting, aggregating DVEmb scores may not capture all nuances of the original algorithm.
>
> >### Q2 Retrain ground truth
>
> We apologize if our explanation of the retraining ground truth was unclear. To clarify,
>
> For most results (Table 2 and shift columns in Table 1), we mimic a standard multi-stage development to train models in sequential stages and stop to save checkpoints at the end of each stage. The next stage begins by loading the checkpoint and re-initializing the momentum and LR scheduler. Retraining removes an entire stage to follow practical development and does not induce artificial optimization failures.
>
> The Insertion and Mislabeled Data cases (column 1&2 in Table 1) correspond to local anomalies rather than distinct stages. We thus don't save/load checkpoints. Instead, we generate the ground truth by interfering with the training loop to skip specific parameter updates. The LR scheduler advances normally for the skipped step, but momentum is not updated since we assume no access to the gradients. This setup serves as a sanity check for fine-grained perturbations.
>
> >### Q3 Runtime and memory
>
> We add experiments to study runtime and memory, and the influence of model/dataset sizes on them. We consider a CV task on CelebA with ResNet18/34 for 2/4/8K data, and an NLP task on CivilComments with LoRA fine-tuning of Pythia70M/160M/410M for 2/4/8K data.
>
> The runtime consists of two parts: Logging (during training) and Attribution (after training). Logging saves optimizer states; the primary time stems from I/O operations. Attribution corresponds to our proposed AA algorithm for perturbation propagation. Below, each row reports the wall-clock time (s) and peak memory for regular training or AA. For AA rows, **Logging/Attribution** time are both reported.
>
> |Model|Mode|Time(2K)|Time(4K)|Time(8K)|Peak Mem|
> |-|-|-|-|-|-|
> |ResNet18|Train|288|460|824|350M|
> ||AA|905/208|1582/406|2903/820|1.1G|
> |ResNet34|Train|299|503|907|512M|
> ||AA|1442/466|2908/850|5538/1610|2G|
>
> |Model(#LoRA Param)|Mode|Time(2K)|Time(4K)|Time(8K)|Peak Mem|
> |-|-|-|-|-|-|
> |Pythia-70M(99K)|LoRA|31|55|104|875M|
> ||AA|36/44|66/101|120/181|1.3G|
> |Pythia-160M(296K)|LoRA|88|164|312|2.4G|
> ||AA|101/129|183/248|346/448|2.4G|
> |Pythia-410M(788K)|LoRA|271|511|979|6.4G|
> ||AA|297/447|547/873|1049/1497|6.4G|
>
> The logging overhead for ResNet is ~3-4x (RN18) and ~5x-6x (RN34) to regular training. For Pythia LoRA, the overhead is ~1.1-1.2x. The overhead mainly comes from I/O and is handled asynchronously. When training computation dominates I/O, it becomes negligible (e.g., LoRA cases). On high-performance machines that clear the I/O bottleneck, this procedure for ResNet can be faster without better GPUs. The attribution time is more manageable; it also scales roughly linearly with model size (e.g., RN34 doubles RN18 time for the same data size), and data size (e.g., 2K->4K->8K; each attribution time doubles for the same model).
>
> The peak memory of AA may be achieved either during logging or attribution, as we use a layer-wise approximation for attribution (App C). If there is one large layer, its attribution may become the peak (ResNet & Pythia-70M). Otherwise, the memory cost of regular training is the peak (Pythia-160M & 410M).
>
> Overall, the runtime and memory are manageable for medium-sized experiments, e.g., from a few to 20-ish minutes. Also, given the scaling trend, we think the method has potential to scale up. While large-scale experiments exceed our current compute and rebuttal time constraints, we acknowledge this in the limitations section and identify structured approximations as future work to bridge this gap.

---

> > ### Author Rebuttal · Reviewer_VT5w · 2026-04-02
> >
> > Technical concerns regarding baselines, ground truth validity, and computational time/efficiency have been addressed by the provided experiments and clarifications.

---

> > > ### Author Response · Authors · 2026-04-06
> > >
> > > We appreciate the reviewer’s support of our paper and the constructive feedback that has helped improve its quality.

---

### Official Review · Reviewer_EWRY · 2026-03-13

**Soundness:** 3
**Presentation:** 3
**Significance:** 3
**Originality:** 3
**Overall Recommendation:** 4
**Confidence:** 2

**Summary:**

This paper addresses a practical concern: modern models typically undergo multiple training stages—pre-training, fine-tuning, adaptation—but when a final model exhibits certain capabilities, biases, or errors, which stage should bear responsibility? The authors formalize this as a counterfactual attribution problem for training stages, proposing an estimation method based on potential outcomes and a first-order approximation to measure a training stage's impact on the final model's behavior. Experiments primarily focus on image classification and text toxicity detection tasks, demonstrating the ability to discern useful signals introduced by specific stages, noisy labels, and spurious correlations.

**Compliance With Llm Reviewing Policy:**

Affirmed.

**Key Questions For Authors:**

Please refer to weaknesses

**Limitations:**

yes

**Strengths And Weaknesses:**

Strengths
1. Assuming each model undergoes multiple training stages, pinpointing when a particular model behavior emerged aids in model localization.
2. Moving beyond static analysis, it expands the attribution scope from individual data points to training stages. This approach is more practical for analyzing real-world development challenges like pretraining, fine-tuning, and domain shift compared to data attribution.

Weaknesses:

1. Replacing training with attribution seems less innovative—the novelty isn't particularly high, though I don't consider this a major flaw.
2. The cost of linear additivity is its inability to distinguish synergistic effects across multiple training segments, which may unfairly penalize multi-stage evaluation training. However, based on my limited theoretical understanding, I'm uncertain if this constitutes a fundamental flaw.
3. Computational complexity is excessively high—could examples be provided to clarify this point?

---

> ### Author Rebuttal · Authors · 2026-03-31
>
> We deeply thank the reviewer for the support of our paper and for the constructive feedback. We address your questions below.
>
> >### Q1 Novelty of replacing training with attribution.
>
> To clarify the novelty, we acknowledge that our method relates to causal responsibility analysis, learning dynamics, and especially data attribution, and we discuss each in our related work. However, our work differs from these lines of research in both the problem formulation and technical approach.
>
> Problem-wise, to our knowledge, we are the first to formally pose the accountability attribution problem, i.e., tracing model behavior to specific stages of a multi-stage training pipeline. We consider it a novel and important problem overlooked in the literature. Compared to some data attribution methods, which also *"replace training with attribution,"* our framework targets the scope of stages instead of individual data points. Meanwhile, we take a model-specific rather than data-centric view. Specifically, traditional data attribution (e.g., influence function-based methods) focuses on the **inherent value** of **individual data points** for their potential to train a model, which corresponds to the "average model" expected from a dataset. In contrast, we target a **specific model** that is already developed and conditioned on its exact training trajectory of **multiple stages**, which naturally handles issues that data-centric methods cannot handle, such as data ordering or repeated samples.
>
> Technically, by conditioning on the specific training trajectory, our method accounts for a more complete optimization process, incorporating learning rate schedules, momentum, and weight decay, which previous data attribution methods overlook, even though some of them can also avoid retraining.
>
> We thank the reviewer for acknowledging that this is not considered a major flaw. We will sharpen the framing of our novelty claims in the revision to make these distinctions more prominent.
>
> >### Q2 Linear additivity and synergistic effects
>
> We agree that linear additivity is insufficient for capturing synergistic effects, but we argue that this doesn't constitute a fundamental flaw.
>
> Theoretically, linear additivity arises naturally from the first-order Taylor expansion underlying our estimators, which is a standard and well-justified approximation in the attribution literature. We acknowledge that one could, in principle, model synergistic effects using a Shapley-like framework. However, such an approach involves computation over the **power set of all training steps**, which is practically infeasible. Our goal is instead to provide a practical tool for auditing real-world models, where additive first-order approximations strike a favorable balance between accuracy and scalability. Furthermore, our error bound (Theorem 4.1) formalizes when this approximation is reliable, which remains small in practical training regimes with small learning rates and well-behaved loss landscapes. Empirically, our method achieves high correlation against retraining-based ground truth across both CV and NLP tasks in multi-stage settings, suggesting that additive stage effects capture the dominant signal.
>
> We nonetheless agree that capturing synergistic effects is a meaningful direction for future work, and we will explicitly highlight this in our limitations section in the revision.
>
> >### Q3 Computational complexity
>
> We acknowledge that the theoretical complexity of our **full** estimator is relatively high and can be challenging for large-scale models when applied naively, and this is also the case for many gradient-based data attribution methods. As discussed in App C, we perform a practical layer-wise approximation to reduce the complexity, which is a significant reduction for deep networks with moderate per-layer dimensions. Such an approximation is sufficient for the tractable computation of small to medium-sized models on a single GPU. Below, we provide new experiments to benchmark the runtime of a CV task on CelebA with ResNet, and an NLP task on CivilComments with Pythia. We pick different model sizes and dataset sizes ranging from 2K to 8K. Each row reports the wall-clock time (s) for attribution computation.
>
> |Model|2K|4K|8K|
> |-|-|-|-|
> |ResNet18|208|406|820|
> |ResNet34|466|850|1610|
>
> |Model (#LoRA Param)|2K|4K|8K|
> |-|-|-|-|
> |Pythia-70M(99K)|44|101|181|
> |Pythia-160M(296K)|129|248|448|
> |Pythia-410M(788K)|447|873|1497|
>
> From these experiments, we observe that 1. the runtime is manageable for handling these medium-sized experiments, e.g., from a few to 20-ish minutes. 2. The attribution time scales roughly linearly with model size (e.g., RN34 doubles RN18 time for the same data size), and data size (e.g., 2K->4K->8K; each attribution time doubles for the same model).

---

> > ### Author Rebuttal · Reviewer_EWRY · 2026-04-03
> >
> > Thank you for the rebuttal. It partially addressed my concerns, particularly by clarifying the trade-off behind the additive approximation and providing concrete runtime results, although I still think cross-stage synergistic effects remain insufficiently explored. Overall, I am satisfied with the response and am happy to maintain my score.

---

> > > ### Author Response · Authors · 2026-04-06
> > >
> > > We appreciate the reviewer's feedback. We agree that capturing non-linear cross-stage synergistic effects is an interesting consideration and a limitation of our current approach. However, rigorously modeling these complex higher-order interactions presents significant theoretical and computational challenges that are out of our current scope. To the best of our knowledge, we do not know of any related work with a similar scope that has explored this direction, and we consider it a compelling direction for future work.

---

### Decision · Program_Chairs · 2026-04-30

**Decision:**

Accept (regular)

**Comment:**

This paper received overall positive evaluations, with two Weak Accept and two Accept recommendations.

The reviewers generally acknowledge the novelty of the proposed method and appreciate the idea of targeting accountability by tracing back to specific stages of training, via a counterfactual attribution framework. The paper is overall recognized as clear and well-written, and the main scientific problem is evaluated as relevant, and effectively described and presented.

On the other hand, reviewers raised some concerns, including empirical evaluation limited to controlled and moderate-scale settings, missing quantitative comparisons with benchmark process-based data attribution methods, and doubts on scalability, due to potential high computational cost, requiring more clarifications.

In their rebuttal, the authors provided additional experiments and specifications, including clarifications on computational cost and quantitative comparisons.

The reviewers generally acknowledged and appreciated the details and specifications provided in the rebuttal, with most of the initial concerns considered resolved, as reflected by improved or maintained positive recommendations, although some limitations remain, concerning scalability and real-world applicability.

After reading the paper, and carefully evaluating the initial reviews, the rebuttal and discussion with the authors, the AC recognizes the value of this work, and agrees with the identified strengths, especially in terms of the significance of the posed research questions and the originality of the proposed approach.

Therefore, the paper is recommended for acceptance, provided that the committed changes are included in the camera-ready version of this work.